# Confidence is Not Universal:
# Task-Dependent Calibration and Emergent Behavior in LLMs

Chaeyun Jang [1]  Moonseok Choi [1]  Yegon Kim [1]  Seungyoo Lee [1]  Juho Lee [1 †]  Hyungi Lee [2 †]

## Abstract

Large language models (LLMs) increasingly support human decision-making, rendering human-interpretable confidence essential. However, it remains unclear whether verbalized confidence calibration generalizes across heterogeneous tasks without degrading accuracy. We show that *universal confidence calibration fails*. Across diverse benchmarks, we identify two incompatible task families with distinct confidence semantics. In reasoning-centric tasks, confidence supervision transfers within the family, often improving calibration while preserving or even improving accuracy, and induces emergent behaviors such as confidence-dependent reasoning length and self-verification. Retrieval- and copy-oriented tasks also exhibit within-family transfer, but fail to generalize to reasoning tasks, with cross-family supervision degrading both calibration and accuracy. Motivated by this finding, we disentangle confidence into reasoning uncertainty and evidence localization uncertainty. This simple decomposition restores cross-family generalization using supervised fine-tuning alone, suggesting that effective confidence alignment requires task-aware semantics rather than a universal scalar notion.

## 1. Introduction

Large language models (LLMs) have now advanced beyond simple natural language generation to demonstrate expert-level performance in domains such as mathematics and coding, increasingly serving as partners that reason, communicate, and support decision-making with users (Achiam et al., 2023; Zhou et al., 2024; Shao et al., 2024). In such in-

teractive contexts, the importance of verbalized confidence becomes particularly pronounced: model outputs must not only provide correct answers but also explicitly express their own confidence so that users can appropriately interpret and act upon them. However, existing LLMs often generate incorrect outputs with high confidence, and such overconfidence can lead to serious decision-making errors if left undetected (Tian et al., 2023; Sahoo et al., 2024; Kim et al., 2025). Consequently, enabling models to convey their confidence in a human-interpretable form (i.e., *verbalized confidence*) has emerged as a critical research direction.

Recent studies have attempted to align verbalized confidence using supervised fine-tuning (SFT) and reinforcement learning (RL) techniques (Lin et al., 2022; Band et al., 2024; Damani et al., 2025; Stangel et al., 2025). While these approaches show promise and some works evaluate across multiple benchmarks, the reported gains often come with non-negligible trade-offs, such as degraded task accuracy under distribution or task shifts, leaving unclear whether confidence alignment is truly *generalizable* in the sense of **improving calibration without sacrificing accuracy**. Moreover, beyond calibration metrics, relatively few studies provide a systematic analysis of how confidence alignment reshapes model behavior; for instance, it remains underexplored when and under what conditions alignment induces accuracy shifts or other unexpected behavioral changes in current models. We summarize representative prior work and their optimization objectives in Table 1.

Thus, the central question of this paper is:

> *Can verbalized confidence in LLMs be consistently calibrated across heterogeneous tasks? If not, under what conditions does such calibration generalize, and when does it break down?*

To investigate this question, we introduce a simple confidence supervision protocol, which we refer to as Confidence-Supervised Fine-Tuning (CSFT), that uses synthesized confidence labels derived from a Monte Carlo estimate of self-consistency. CSFT is deliberately designed to be an experimental instrument rather than a task-specific optimization method, enabling us to isolate and analyze the effects of confidence calibration itself while minimizing interference from

---

[†]Corresponding authors. [1]Korea Advanced Institute of Science and Technology, Republic of Korea [2]Kookmin University, Republic of Korea. Correspondence to: Juho Lee <juholee@kaist.ac.kr>, Hyungi Lee <lhk2708@kookmin.ac.kr>.

*Proceedings of the $43^{rd}$ International Conference on Machine Learning*, Seoul, South Korea. PMLR 306, 2026. Copyright 2026 by the author(s).

architectural choices or reinforcement learning dynamics.

Our central finding is that **universal confidence calibration systematically fails**. Across a broad range of tasks, we identify two orthogonal axes, or task families, across which confidence calibration does not generalize. In the first, reasoning-centric family, the model primarily relies on its internal parametric knowledge (*Cluster 1*; including GSM8K (Cobbe et al., 2021), MATH (Hendrycks et al., 2021), HellaSwag (Zellers et al., 2019), ARC (Clark et al., 2018), and MMLU (Wang et al., 2024)). Confidence supervision transfers effectively within the family, often improving calibration while maintaining or sometimes even improving accuracy. Upon closer inspection, CSFT induces two systematic effects within this family: 1) an emergent coupling between confidence and reasoning behavior, where low confidence triggers longer, self-verifying reasoning traces, and 2) a smoothing effect in the distribution of numeric confidence tokens. Concrete examples of these effects are provided in Figure 1 and Figure 3. Despite these within-family gains, models trained under this regime exhibit substantial degradation when evaluated on tasks outside the family. Retrieval- and copy-oriented tasks such as NIAH in RULER (Hsieh et al., 2024) (*Cluster 2*) exhibit a similar, reversed pattern: confidence supervision transfers within the family but fails to generalize to reasoning-centric tasks, and the emergent behavioral effects observed in Cluster 1 do not arise. This suggests that verbalized confidence in retrieval settings reflects a fundamentally different notion of uncertainty, due to the existence and localization of external evidence rather than internal multi-step reasoning. Detailed descriptions of the datasets and representative examples are provided in Figure 2.

Motivated by this diagnosis, we propose a simple remedy. We disentangle verbalized confidence according to its semantic role and train the resulting components jointly using a prompting-based output structure. Specifically, we elicit and supervise two confidence values, capturing uncertainty in reasoning (Cluster 1) and in retrieval/copy (Cluster 2), respectively. This decomposition restores compatibility across the two task families and enables generalization to an additional held-out family represented by ContractNLI (Koreeda & Manning, 2021). At the same time, the emergent behavioral signatures associated with each axis are preserved where they are expected to arise. Notably, these gains are achieved using SFT alone, without relying on RL-based calibration alignment.

Overall, our contributions are:

- We study the generalization of verbalized confidence calibration across heterogeneous LLM tasks under controlled supervision.

- We identify incompatible confidence axes, showing robust within-family transfer and cross-family failure.

- We reveal axis-specific emergent effects of confidence supervision, including reasoning-length modulation and confidence-token smoothing.

- We propose a simple disentangled verbalized confidence scheme that restores cross-family generalization without reinforcement learning.

## 2. Confidence Supervision Setup

We adopt Confidence-Supervised Fine-Tuning (CSFT) as a simple and controlled setup for supervising verbalized confidence in LLMs. Rather than proposing a method optimized for task performance, we use CSFT as an experimental instrument to study how explicitly supervising verbalized confidence influences model behavior. Under this setup, the model produces either a standard Chain-of-Thought (CoT) rationale followed by a final answer, or only a final answer in the answer-only setting, and is trained solely to calibrate its verbalized confidence, without any explicit supervision on the reasoning process or the answer.

Our primary goal is to evaluate generalization that achieves **calibration without degrading the base model's task accuracy**. Here, we define the baseline accuracy as the model's accuracy when it performs the task without being asked to elicit confidence. To ensure a fair and consistent comparison, we fix confidence elicitation to a *suffix prompt* appended after the answer, thereby minimizing the extent to which confidence reporting interferes with the answer generation process.

**Formatting** First, we guide the target LLM to produce both an answer and a corresponding confidence value in a fixed format. Specifically, given a query $q$, the decoder of a pre-trained LLM is prompted to generate a structured response that includes: (i) either a CoT reasoning trace $r$ *followed by* a final answer $a$, enclosed within `<think>` ... `</think>` and `<answer>` ... `</answer>` tags, respectively, *or* only the final answer $a$ in the answer-only setting; followed by (ii) a suffix confidence prompt (see § D), which elicits a discrete confidence score $c \in 0, 10, \ldots, 100$ expressed in `<confidence>` ... `</confidence>` tags. During training, only the confidence score $c$ is directly supervised, while the reasoning $r$ (when present) and final answer $a$ remain unconstrained.

### 2.1. Training Procedure

In this section, we describe the training procedure of CSFT. Our goal is to efficiently endow the model with the ability to produce calibrated verbalized confidence, while preserving

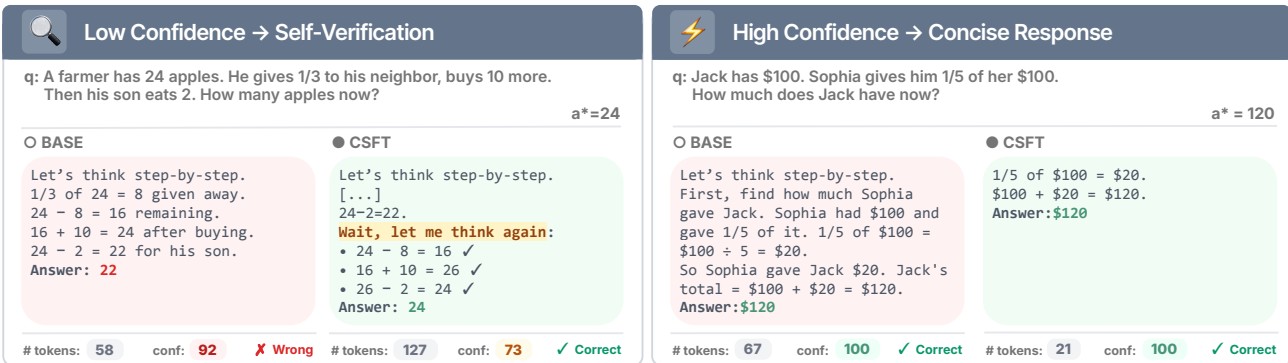

Figure 1. **Confidence reshapes reasoning behavior.** Left: low confidence triggers self-verification, leading to correct answers despite the errors. Right: high confidence yields concise response. CSFT induces an emergent coupling between confidence and generation length.

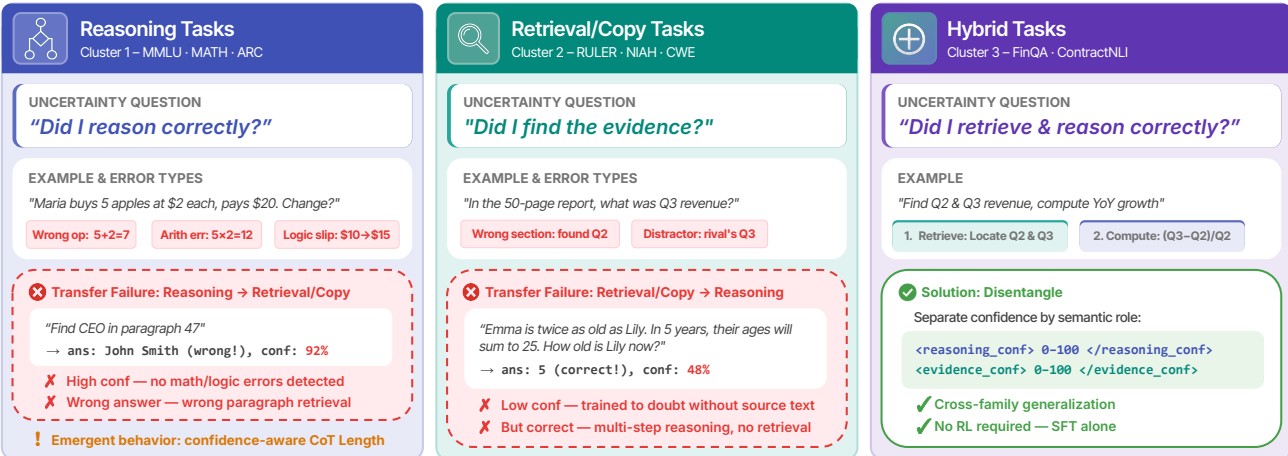

Figure 2. **Taxonomy of verbalized confidence across task families.** Reasoning and retrieval tasks exhibit incompatible confidence semantics: models trained on one cluster miscalibrate on the other, and both cannot generalize to hybrid task calibration. We propose disentangling confidence into reasoning and retrieval/copy, which restores cross-family generalization via SFT alone.

as many of the pre-trained LLM's learned features as possible. To achieve this, we fine-tune the base model $f_{\theta_0}$ using the Low-Rank Adaptation (LoRA; Hu et al., 2022) method. Here, $\theta_0$ denotes the fixed base model's parameters, and $\theta$ represents these fixed parameters augmented with additional learnable LoRA weights.

**Self-Confidence Label Generation.** As outlined in (ii) of the above **Formatting** paragraph, our objective is to ensure that the model generates a discrete, well-calibrated confidence score within the prescribed format. To guide the model toward producing such confidence scores, we directly constructed confidence labels for each query $q$ in the training data. Here, we generate the confidence labels for the training data using the base model $f_{\theta_0}$. Because the labels are prepared before training, they can be reused during training, minimizing computational cost.

In order to compute the confidence label, we first sample

$K$ full generations $\{(r_0^{(i)}, a_0^{(i)})\}_{i=1}^{K} \sim f_{\theta_0}(\cdot \mid q)$ before training and estimate the empirical accuracy as

$$\hat{p}(q) = \frac{1}{K} \sum_{i=1}^{K} \mathbb{1}\big[a_0^{(i)} = a^{\star}\big],$$

where $a^{\star}$ denotes the gold answer. The reason for generating $K$ answers is to examine the distribution formed when the LLM model is queried $K$ times with the same query, thereby assessing how confident the model is about its answers. Since we use model weights $\theta_0$ to generate responses, the dataset can be constructed in advance, and the same labels can be used consistently throughout training, enabling efficient learning. This setup is feasible because, during fine-tuning, we adopt the LoRA method, which preserves the pretrained features while adding the ability to output well-calibrated confidence scores. Consequently, even after training, the model's confidence remains similar to that of the base model. Further, we provide additional results using adaptive label rather than fixed ones in § C.2.

When constructing the label, we first parsed the text between the `<answer>` ... `</answer>` tags in the LLM output to extract each $a_0^{(i)}$, and then checked whether it exactly matched the gold answer $a^\star$. Finally, the self-confidence label was obtained by discretizing the empirical accuracy as

$$c = 10 \cdot \lfloor 10 \cdot \hat{p}(q) \rfloor,$$

which maps $\hat{p}(q) \in [0,1]$ to a confidence value $c \in \{0, 10, \ldots, 100\}$.

**Training Objective.** Our objective is to ensure that the target LLM produces well-calibrated confidence scores. Therefore, during training, we basically do not apply loss to the reasoning trace $r$ or the final answer $a$; instead, we compute the loss solely on the confidence score. Specifically, let $T_c$ denote the token positions corresponding to the entire confidence span, including the `<confidence>` ... `</confidence>` tags. CSFT minimizes the masked cross-entropy loss over these positions:

$$\mathcal{L}_{\text{CSFT}} = -\sum_{t \in T_c} \log p_\theta(y_t \mid y_{<t}, q),$$

where $p_\theta(y)$ denotes the predicted probability of token $y$ under the LLM $f_\theta$.

# 3. Empirical Findings

## 3.1. Experimental Design

**Datasets.** We consider a diverse set of reasoning and knowledge-intensive benchmarks. For confidence training, we use GSM8K (Cobbe et al., 2021) and RULER-4K (Hsieh et al., 2024) as training tasks. For evaluation, we assess generalization across a broader set of tasks, including RULER-4K, RULER-8K, MMLU (Wang et al., 2024), MATH (Hendrycks et al., 2021), and GSM8K (Cobbe et al., 2021). Further details of the training and evaluation datasets are provided in § B.

**Baselines and Training.** All models are initialized from LLAMA-3.2-3B-INSTRUCT (Grattafiori et al., 2024). We fine-tune each model for 2,000 optimization steps under a range of hyperparameter configurations, and select the checkpoint with the lowest validation loss. Complete training details and hyperparameter settings are provided in § B.

Beyond CSFT, we also evaluate the generalization behavior of representative RL-based confidence alignment methods. As summarized in Table 1, prior work either (i) does not report out-of-distribution (OOD) evaluations, or (ii) fails to demonstrate the type of generalization we study-*i.e.*, achieving improved calibration without degrading the base model's accuracy. We compare CSFT against two major families of RL approaches: (i) log-loss RL and (ii) Brier-score RL. For

a controlled comparison, both RL baselines are optimized using Group Relative Policy Optimization (GRPO) (Shao et al., 2024) without standard deviation normalization.

Importantly, prior studies report that confidence-only rewards can lead to unstable optimization or degenerate behaviors (Damani et al., 2025). Therefore, all RL baselines additionally incorporate a correctness reward, following standard practice in the literature, to stabilize training. As a result, RL-based approaches operate under a strictly more favorable *in-domain* training setting, as they benefit from direct task-level supervision in addition to confidence alignment.

**Evaluation.** We evaluate all models using standard accuracy metrics, as well as confidence-aware measures including area under the receiver operating characteristic curve (AUROC) (Hendrycks & Gimpel, 2016), Expected Calibration Error (ECE) (Naeini et al., 2015), and the Brier Score (BS) (Brier, 1950). Throughout the paper, we collectively refer to AUROC, ECE, and BS as **calibration metrics**. Formal definitions and implementation details for each metric are provided in § B.1.1.

## 3.2. Cross-Task Generalization Reveals Two Task Families of Confidence Semantics

We evaluate whether confidence supervision on a source task generalizes to unseen target tasks. Concretely, we apply confidence supervision on a source task and measure both accuracy and calibration on held-out target tasks.

We find that cross-task generalization is highly structured, revealing two task families with incompatible confidence semantics. We group tasks when confidence supervision consistently preserves or improves both accuracy and calibration under transfer, and separate them otherwise. **Cluster 1** consists of GSM8K, MATH, and MMLU, where verbalized confidence primarily reflects **reasoning** stability and internal consistency. **Cluster 2** consists of long-context **retrieval and copy** tasks from RULER (e.g., RULER-4K and RULER-8K), including NIAH, CWE, FWE, and SQUAD, where confidence is primarily tied to whether the model successfully localizes the relevant evidence.

As shown in Table 4, confidence supervision on GSM8K transfers reliably within Cluster 1, but leads to severe miscalibration when applied to Cluster 2; this failure persists even for RL-based baselines that include explicit correctness rewards. Conversely, supervision on Cluster 2 does not yield consistent improvements on Cluster 1, suggesting that the induced confidence semantics do not align with reasoning-centric tasks.

Overall, these results indicate that confidence supervision does not induce a task-agnostic notion of confidence. In-

*Table 1.* **Methodological comparison of verbalized confidence tuning.** CE: cross-entropy; LS: log score; BS: Brier score. More detailed descriptions of these methods are provided in § A.

| Method | Optimization | Objective | Conf. Supervision | OOD Gen. | Conf. Dim. |
|---|---|---|---|---|---|
| Lin et al. (2022) | SFT | $\min_\theta \mathbb{E}\big[\mathrm{CE}(\hat{p}_\theta,\, p^*)\big]$ | Empirical acc. per sub-task | ✕ | Single |
| Band et al. (2024) | SFT + PPO | $\max_\theta \mathbb{E}\big[\mathrm{LS}(\hat{p}_\theta,\, y)\big]$ | Self-consistency / Correctness ($y$) | ✕ | Single |
| Stangel et al. (2025) | PPO | $\max_\theta \mathbb{E}\big[\mathrm{LS}(\hat{p}_\theta,\, y)\big]$ | Correctness ($y$) | ✕ | Single |
| Damani et al. (2025) | GRPO | $\max_\theta \mathbb{E}\big[-\mathrm{BS}(\hat{p}_\theta,\, y) + \lambda y\big]$ | Correctness ($y$) | △ | Single |
| **Ours** | SFT | $\min_\theta \mathbb{E}\big[\mathrm{CE}(\hat{p}_\theta^r,\, p^{r*}) + \mathrm{CE}(\hat{p}_\theta^e,\, p^{e*})\big]$ | MC-estimated $P(\text{correct})$ | ✓ | Multi |

*Table 2.* **Evaluation results reported as cluster-level averages over reasoning tasks (Cluster 1) and retrieval/copy tasks (Cluster 2).** Within each cluster, metrics are aggregated using weighted macro-averaging according to the task-specific data proportions. *Cell colors indicate relative change from the Base model within each cluster: green (improved), red (degraded), and white (no change).*

| | Cluster 1 | | | | Cluster 2 | | | |
|---|---|---|---|---|---|---|---|---|
| Method | Acc. ↑ | ECE ↓ | Brier ↓ | AUROC ↑ | Acc. ↑ | ECE ↓ | Brier ↓ | AUROC ↑ |
| Base | 0.6163 | 0.3390 | 0.3589 | 0.5003 | 0.6954 | 0.2745 | 0.3824 | 0.6256 |
| Log Score RL on GSM8k | 0.6540 | 0.2125 | 0.2433 | 0.5344 | 0.6062 | 0.3832 | 0.3818 | 0.5405 |
| Log Score RL on RULER-4k | 0.5509 | 0.4095 | 0.4158 | 0.4443 | 0.6954 | 0.2214 | 0.2167 | 0.6962 |
| Brier RL on GSM8k | 0.6368 | 0.2079 | 0.2465 | 0.5312 | 0.6854 | 0.2837 | 0.2789 | 0.6898 |
| Brier RL on RULER-4k | 0.5873 | 0.3952 | 0.4017 | 0.5262 | 0.6954 | 0.2168 | 0.2225 | 0.7253 |
| CSFT on GSM8k | 0.6282 | 0.1086 | 0.1684 | 0.7351 | 0.6976 | 0.6375 | 0.6461 | 0.4838 |
| CSFT on RULER-4k | 0.6001 | 0.4288 | 0.4196 | 0.5751 | 0.7000 | 0.1111 | 0.1106 | 0.9131 |

stead, optimizing confidence for one task family implicitly enforces inductive biases that conflict with the other, explaining the observed breakdown in cross-task generalization. Therefore, confidence should be viewed as a task-family-dependent construct rather than a universal scalar.

**The failure is not an artifact of LoRA.** A natural concern is whether the observed cross-family breakdown stems from the limited capacity of LoRA rather than from confidence semantics. To rule this out, we repeat the single-task CSFT experiments under *full-parameter* fine-tuning, which updates all model weights (Table 3). The same asymmetric pattern persists: within-family transfer (GSM8K→GSM8K, RULER-4K→RULER-4K) yields strong calibration, while transferring reasoning supervision to retrieval (GSM8K→RULER-4K) severely degrades calibration (ECE 0.60, AUROC near chance), consistent with the LoRA results. Because full fine-tuning has strictly greater capacity than LoRA yet reproduces the identical failure, the breakdown reflects a fundamental incompatibility of confidence semantics across task families rather than an optimization or capacity artifact.

### 3.3. Emergent Behaviors under Confidence Alignment

Unlike RL-based baselines, CSFT does not provide explicit supervision of answer correctness or CoT trace. Nevertheless, we observe a substantial *gain* in accuracy on the GSM8K evaluation set when training on GSM8K, improving from 69% to 75%. Beyond calibration, CSFT induces additional emergent behaviors that are absent when training

*Table 3.* **Cross-family failure persists under full-parameter fine-tuning** (LLAMA-3.2-3B-INSTRUCT, single-task CSFT). Within-family transfer remains well-calibrated, while transferring reasoning supervision to retrieval collapses, mirroring the LoRA results.

| Train → Test | Acc. ↑ | ECE ↓ | Brier ↓ | AUROC ↑ |
|---|---|---|---|---|
| GSM8K → GSM8K (within) | 0.7081 | **0.1324** | **0.1912** | **0.7460** |
| GSM8K → RULER-4K (cross) | 0.7277 | 0.5958 | 0.5960 | 0.4781 |
| RULER-4K → RULER-4K (within) | 0.7277 | **0.0760** | **0.0828** | **0.9397** |
| RULER-4K → GSM8K (cross) | 0.6755 | 0.1946 | 0.2320 | 0.6868 |

on Cluster 2, further supporting that the two clusters correspond to different confidence semantics. We analyze two dominant phenomena: (i) number token smoothing and (ii) CoT length optimization with self-verification.

**Number Token Smoothing** When trained on Cluster 1, CSFT exhibits a local smoothing effect over discrete confidence number tokens despite using standard cross-entropy (CE) loss. A naive CE objective concentrates probability mass on the exact target token, which can ignore semantic continuity among nearby confidence values (e.g., '60', '70', '80'). Since verbalized confidence is expressed as discrete token sequences, overly sharp token distributions hinder credit assignment to semantically adjacent numbers, weakening the alignment between predicted confidence and actual correctness and increasing calibration error (Huang et al., 2025).

CSFT partially mitigates this issue in Cluster 1. As shown in Figure 3a, when generating a confidence token, the model allocates non-trivial probability mass to nearby numeric

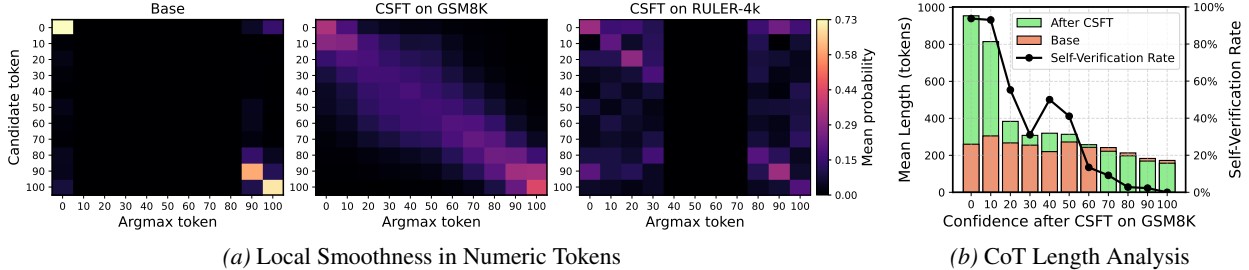

*(a)* Local Smoothness in Numeric Tokens        *(b)* CoT Length Analysis

*Figure 3.* **Emergent behaviors induced by CSFT on Cluster 1 (GSM8K).** (a) Token-level probability distributions exhibit *local smoothness over numeric tokens* when CSFT is trained on GSM8K: predicting a confidence number allocates non-trivial probability mass to nearby numeric tokens. This effect is not observed when training CSFT on Cluster 2 (RULER-4K). (b) Output length and self-verification frequency as a function of confidence for CSFT trained on Cluster 1: lower-confidence predictions tend to include explicit self-verification behaviors, resulting in longer responses, whereas higher-confidence predictions remain concise.

tokens, forming a soft neighborhood in the number space. This behavior induces a smoother confidence representation and improves calibration without modifying the loss or architecture. In contrast, this smoothing pattern is not observed when CSFT is trained on Cluster 2, suggesting that Cluster 2 confidence is not organized around an ordinal number-space semantics but is instead tied to likelihood-driven behaviors.

**CoT Length Optimization and Self-Verification** We next analyze an emergent behavior induced by CSFT in Cluster 1: low-confidence outputs are accompanied by longer CoT traces and explicit self-verification. Figure 3b reports (1) the average CoT length as a function of predicted confidence and (2) the proportion of answers exhibiting self-verification across confidence bins.

We observe a clear monotonic trend: as confidence decreases, CSFT-trained models generate substantially longer CoT traces. Importantly, this effect is not merely an artifact of task difficulty. Low-confidence bins do correspond to inherently harder questions, for which the pre-trained baseline also tends to produce longer answers; Figure 3b therefore compares CSFT against the Base model on the *same set of questions within each confidence bin*. Even under this difficulty-controlled comparison, CSFT yields a residual length increase of roughly 40–50% over the Base model in the lowest confidence bins, indicating genuine confidence-aware length modulation beyond what task difficulty alone explains. This residual increase correlates with a high self-verification rate, and corrected answers almost always contain explicit verification steps. Self-verification is identified by prompting GPT-4.1-MINI (Achiam et al., 2023) to detect explicit verification behaviors from generated CoT traces (see Figure 19). These patterns generalize beyond the training distribution and emerge similarly on held-out reasoning tasks (§ C.4).

## 4. Disentangled Verbalized Confidence

Motivated by our empirical findings, we introduce a disentangled formulation of verbalized confidence that separates distinct uncertainty semantics. Specifically, we decompose confidence into *reasoning confidence* and *evidence confidence*, addressing the interference observed under scalar confidence supervision. This disentanglement is introduced as a minimal structural remedy rather than a new optimization method.

**Formatting.** Given a query $q$, the model generates a CoT reasoning trace $r$ and a final answer $a$ as before, followed by two confidence values: reasoning confidence $c_r$ and evidence confidence $c_e$, expressed in `<reasoning_confidence>` and `<evidence_confidence>` tags. Both confidence values are discretized as $c_r, c_e \in \{0, 10, \ldots, 100\}$. During training, only the confidence tokens are supervised; the reasoning trace $r$ and answer $a$ remain unconstrained.

**Confidence Labels.** Confidence labels are constructed via Monte Carlo self-consistency, estimating the empirical probability of correctness $\hat{p}_r(q)$ as described in § 2. For Cluster 1, we set the evidence confidence label to `N/A`, whereas for Cluster 2, we set the reasoning confidence label to `N/A`. The detailed prompting templates are provided in Figure 13.

**Training Objective.** Let $T_{c_r}$ and $T_{c_e}$ denote the token positions corresponding to the reasoning and evidence confidence spans. We apply masked cross-entropy loss independently to each component:

$$\mathcal{L}_{\text{DVC}} = -\sum_{t \in T_{c_r}} \log p_\theta(y_t \mid y_{<t}, q)$$
$$-\sum_{t \in T_{c_e}} \log p_\theta(y_t \mid y_{<t}, q),$$

where $p_\theta(y)$ denotes the predicted probability under $f_\theta$. No loss is applied to the reasoning or answer tokens.

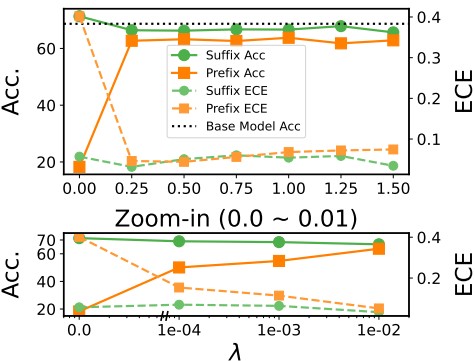

*Figure 4.* **Ablation analysis of CSFT using LLaMA-3.2-3B trained on GSM8K.** Effect of KL divergence and confidence prompt position during training.

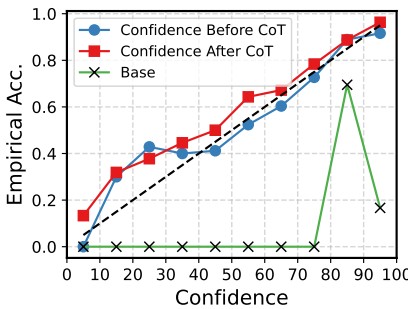

*Figure 5.* **Ablation analysis of CSFT using LLaMA-3.2-3B trained on GSM8K.** Effect of placing the confidence query before CoT and answer generation.

## 5. Main Experiments

### 5.1. LLaMA-3.2-3B-Instruct

We first conduct our main experiments using LLAMA-3.2-3B-INSTRUCT. Unless otherwise specified, all confidence-aligned models, including both our method and competing baselines, are trained in a multi-task setting. When the training data is not explicitly stated, the training mixture consists of GSM8K (Cluster 1) and RULER-4K (Cluster 2). The key distinction is that only our method learns *disentangled confidence supervision*, whereas all baselines are trained to output a single scalar confidence.

**Evaluation protocol.** To evaluate cross-task generalization, we report results on unseen datasets that are excluded from training. Concretely, we evaluate on MATH (Cluster 1) and RULER-8K (Cluster 2), which are held out from the multi-task training mixture. Moreover, to test generalization to hybrid uncertainty regimes, we additionally evaluate on CONTRACTNLI (Cluster 3), which exhibits both reasoning-style and evidence-localization. This setup intentionally probes whether a confidence supervision signal

learned from one task family can remain meaningful under a different family and under mixed-family conditions.

**Baselines.** We compare against task-level confidence supervision, adapted from Lin et al. (2022) (see Table 1). This baseline assigns each example a task-level empirical accuracy as the confidence label, estimated from held-out performance statistics. Because it requires explicit task/-subtask labels, this method can only be applied to datasets where such annotations are available. Thus, as an exception, we train this baseline using MATH and RULER-4K (rather than GSM8K and RULER-4K), which implies that for this baseline, MATH is partially in-domain at evaluation time.

**Results.** Table 4 shows that our method achieves robust performance across all three evaluation datasets. In particular, while CSFT trained on RULER-4K can yield substantial gains on the retrieval benchmark, it fails to transfer to MATH and degrades on the hybrid benchmark CON-TRACTNLI. Conversely, CSFT trained on GSM8K transfers well within Cluster 1 but deteriorates under Cluster-2 evaluation. These results highlight that a *single scalar confidence* is an incompatible supervision target across heterogeneous task families, whereas **disentangling confidence** enables consistent cross-family generalization and stability.

**Generalization to entirely different task paradigms.** To test whether our taxonomy extends beyond the reasoning/retrieval benchmarks used for diagnosis, we evaluate on two tasks from qualitatively different paradigms: XSTEST (Röttger et al., 2024), which probes safety/refusal boundaries, and HALUEVAL (Li et al., 2023), which probes hallucination detection. Both require judging answerability or factual correctness from the model's internal parametric knowledge rather than localizing external evidence, and thus fall naturally within Cluster 1. As shown in Table 6, our method (trained on the GSM8K+RULER-4K mixture) consistently improves over the Base model across all metrics on both tasks, with large AUROC and ECE gains. This confirms that tasks from seemingly "out-of-paradigm" domains in fact converge within our existing cluster structure, indicating that the framework already subsumes a substantially broader range of real-world uncertainty than the diagnostic benchmarks alone suggest.

### 5.2. Qwen-3-8B

We next validate the generality of our findings using a stronger and architecturally distinct base model, QWEN-3-8B (Yang et al., 2025). We train confidence-aligned models on a multi-task mixture of MATH and RULER-4K, representing reasoning-centric and retrieval-centric supervision signals, respectively.

*Table 4.* **Out-of-distribution evaluation on held-out benchmarks.** Task-level CSFT, trained in-domain on each target dataset, serves as a naive baseline. We examine that CSFT trained on GSM8K/RULER-4k generalizes to MATH/RULER-8k but fails on RULER-8K/MATH. Neither single-cluster model transfers to the ContractNLI. Our method, trained on both clusters, achieves robust generalization across all three task families, including the unseen hybrid benchmark.

| Method | MATH | | | | RULER-8k | | | | ContractNLI | | | |
|---|---|---|---|---|---|---|---|---|---|---|---|---|
| | Acc. ↑ | ECE ↓ | Brier ↓ | AUROC ↑ | Acc. ↑ | ECE ↓ | Brier ↓ | AUROC ↑ | Acc. ↑ | ECE ↓ | Brier ↓ | AUROC ↑ |
| Base | 0.3504 | 0.5871 | 0.5784 | 0.5290 | 0.6677 | 0.2908 | 0.2964 | 0.6342 | 0.4308 | 0.5021 | 0.5006 | 0.5103 |
| Task-level CSFT* | 0.3502 | 0.2042 | 0.2216 | 0.6882 | 0.6492 | **0.0762** | 0.1222 | **0.9076** | 0.4529 | 0.2423 | 0.2508 | 0.4993 |
| CSFT on GSM8K | 0.3544 | 0.1424 | 0.1788 | 0.8403 | 0.6642 | 0.6351 | 0.6399 | 0.4911 | 0.4725 | 0.1552 | 0.2798 | 0.5421 |
| CSFT on RULER-4k | 0.3428 | 0.3392 | 0.3398 | 0.5404 | **0.6738** | 0.1155 | **0.1186** | 0.9014 | 0.4295 | 0.1996 | **0.2430** | 0.5551 |
| Ours | **0.3776** | **0.1162** | **0.1681** | **0.8498** | 0.6700 | 0.1466 | 0.1522 | 0.8263 | **0.4907** | **0.1181** | 0.2527 | **0.6089** |

*Table 5.* **Cluster-level evaluation on Qwen-3-8B trained on MATH and RULER-4K.** Results are macro-averaged within each cluster: Cluster 1 - reasoning, Cluster 2 - retrieval, and Cluster 3 - hybrid. Our disentangled confidence method consistently improves accuracy and calibration over the base model across all clusters, confirming that task-aware confidence decomposition generalizes to stronger models.

| Cluster | Method | Acc. ↑ | ECE ↓ | Brier ↓ | AUROC ↑ |
|---|---|---|---|---|---|
| **Cluster 1** | Base | 0.6530 | 0.2833 | 0.2707 | 0.8260 |
| | Ours | **0.6745** | **0.1317** | **0.1364** | **0.8263** |
| **Cluster 2** | Base | 0.8531 | 0.1326 | 0.1339 | 0.7735 |
| | Ours | **0.8685** | **0.1100** | **0.1198** | **0.7986** |
| **Cluster 3** | Base | 0.5328 | 0.3895 | 0.3934 | 0.6194 |
| | Ours | **0.5973** | **0.2857** | **0.3104** | **0.6304** |

*Table 6.* **Generalization to different task paradigms** (LLAMA-3.2-3B-INSTRUCT). XSTEST ($N$=450; safety/refusal) and HALUEVAL ($N$=4507, general split; hallucination detection) both exhibit Cluster 1-like transfer, and our method improves over Base across all metrics.

| Task | Method | Acc. ↑ | ECE ↓ | Brier ↓ | AUROC ↑ |
|---|---|---|---|---|---|
| XSTEST | Base | 0.6756 | 0.5802 | 0.5893 | 0.3814 |
| | Ours | **0.7556** | **0.3653** | **0.3639** | **0.7206** |
| HALUEVAL | Base | 0.5891 | 0.3050 | 0.3361 | 0.4958 |
| | Ours | **0.6676** | **0.2793** | **0.2788** | **0.6098** |

**Cluster-level evaluation.** To summarize generalization behavior at the family level, we report macro-averaged performance across three clusters: **Cluster 1** (reasoning tasks), evaluated using GPQA-DIAMOND (Rein et al., 2024); **Cluster 2** (retrieval/copy tasks), evaluated using RULER-4K and RULER-8K; and **Cluster 3** (hybrid uncertainty), evaluated using CONTRACTNLI. For each cluster, we report macro averages of accuracy and calibration metrics (ECE, Brier score, and AUROC).

**Results.** As shown in Table 5, our method consistently improves over the base QWEN-3-8B model across ALL CLUSTERS. Notably, the gains persist not only on reasoning-centric and retrieval-centric benchmarks but also in the hy-

brid regime, confirming that disentangled confidence provides a scalable and model-agnostic mechanism for transferring confidence supervision across heterogeneous uncertainty semantics.

### 5.3. Qualitative Examples.

In § C and Figure 6, we provide qualitative comparisons between the base model and CSFT fine-tuned LLAMA-3.2-3B-INSTRUCT models trained on Cluster 1, Cluster 2, or Cluster 3.

For reasoning-centric tasks, we observe that the model fine-tuned on Cluster 1 produces substantially longer generations under uncertain cases, often exhibiting explicit self-verification behaviors compared to the base model. Importantly, this self-verification enables the CSFT model to correct errors that the base model fails to resolve. We further find that this behavior generalizes to code generation tasks, with examples shown in Figure 9 and Figure 10.

For retrieval-oriented tasks, the model fine-tuned on Cluster 2 tends to output confidence values close to zero when it is incorrect, whereas the model fine-tuned on Cluster 1 often outputs near-zero confidence even for correct predictions. Finally, on tasks that mix retrieval and reasoning components, only the model trained jointly on Cluster 3 achieves consistently reliable calibration.

## 6. Ablation Studies

**KL Regularization and Prompt Position.** We study how the position of the confidence prompt interacts with Kullback-Leibler divergence (KL) regularization. Our main setting does not apply KL regularization to the answer or CoT spans; however, we additionally explore a variant that imposes KL constraints specifically on the CoT and answer tokens. We vary the KL regularization weight $\lambda$, with particular attention to behavior near $\lambda = 0$ (see Figure 4, Zoom-in figure), to determine whether apparent gains at low $\lambda$ are stable or merely artifacts of tuning.

We compare two confidence elicitation strategies (see § D):

*Table 7.* Ablation analysis on the impact of confidence label quality and the inclusion of confidence prompt. Results are reported as differences relative to CSFT on GSM8K using LLAMA3.2-3B-INSTRUCT.

| Variant | ACC | ECE |
|---|---|---|
| w/o Correct label | -2.14 | +0.05 |
| w/o Conf question | Training collapsed | |

a *suffix* setting where the confidence prompt is appended after the model's answer (described in § 2), and a *prefix* variant where the confidence prompt is inserted immediately after the question, before any reasoning begins. Notably, the prefix setting consistently yields lower accuracy than the base model, and performance rapidly deteriorates as KL regularization is removed (i.e., $\lambda \rightarrow 0$). This suggests that when confidence is requested before generation, the model may exploit the added freedom to optimize calibration metrics (e.g., ECE) at the expense of actual reasoning quality, becoming well-calibrated but confidently incorrect.

In contrast, the suffix setting remains stable or even improves in the absence of KL regularization. Since confidence is predicted independently after the full generation, removing the KL constraint does not interfere with the reasoning process and can enable better post hoc alignment of confidence with correctness.

**Effect of Confidence Prompt Position after CSFT.** To evaluate whether the model's confidence relies on observing the length or content of the generated CoT, we compare calibration when confidence is elicited before versus after CoT generation. As shown in Figure 5, the two reliability curves are broadly similar, suggesting that the model does not depend heavily on CoT visibility and instead bases its confidence on internal uncertainty.

**Label Quality and Confidence Prompt** To assess the importance of confidence supervision and prompt design for stable training and reliable performance, we conduct two ablations (Table 7).

First, randomizing the confidence labels within the `<confidence>` tag causes clear degradation across all metrics, confirming that accurate supervision is critical for learning calibrated confidence.

Second, removing the explicit confidence prompt and forcing a scalar confidence after the final answer via a repeated `<answer>` tag leads to complete training collapse, suggesting that explicit prompting is necessary for the model to ground and learn a meaningful confidence signal.

## 7. Conclusion

In this work, we revisit a key assumption in verbalized confidence alignment: that confidence is a universal scalar transferable across tasks. Using controlled CSFT, we show this assumption fails, and instead uncover two task families with incompatible confidence semantics: reasoning-centric tasks, where confidence reflects reasoning stability, and retrieval/copy-oriented tasks, where it tracks evidence localization.

Beyond calibration, confidence supervision can reshape model behavior. On reasoning tasks, it induces an emergent coupling between confidence and generation, including confidence-dependent reasoning length, self-verification, and local smoothing over discrete confidence tokens. These effects are absent under retrieval-based supervision.

Motivated by this diagnosis, we propose a minimal remedy: disentangling verbalized confidence via joint SFT without RL. This simple decomposition restores cross-family compatibility and generalizes reliably to held-out tasks, including hybrid uncertainty regimes such as CONTRACTNLI. Overall, our findings suggest that effective confidence alignment requires task-aware semantics.

## Acknowledgements

This work was supported by Institute of Information & communications Technology Planning & Evaluation(IITP) grant funded by the Korea government(MSIT) (No.RS-2019-II190075, Artificial Intelligence Graduate School Program(KAIST)). This work was supported by Institute of Information & communications Technology Planning & Evaluation(IITP) grant funded by the Korea government(MSIT) (No.2022-0-00184, Development and Study of AI Technologies to Inexpensively Conform to Evolving Policy on Ethics). This work was supported by the National Research Foundation of Korea(NRF) grant funded by the Korea government(MSIT) (RS-2022-NR070855). HG was supported by the Institute of Information & Communications Technology Planning & Evaluation(IITP) grant funded by the Korea government(MSIT) (No.RS-2025-02219317, AI Star Fellowship(Kookmin University)) and the National Research Foundation of Korea(NRF) grant funded by the Korea government(MSIT) (RS-2026-25480825).

## Impact Statement

This paper presents work whose goal is to advance the field of machine learning. Our work aims to improve the reliability and interpretability of LLM confidence, which may help users better assess when model outputs are trustworthy. While improved confidence signals can support safer deployment, they may also increase over-reliance or be misused to make incorrect outputs appear more credible. We encourage careful evaluation under distribution shifts and the use of confidence alongside additional safeguards such as verification and human oversight.

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

# A. Related Works

### A.1. Alignment of Verbalized Confidence in LLMs

Recently, a growing body of work has sought to train LLMs to explicitly express confidence in their own outputs. Early efforts in this direction supervised confidence by assigning task-level empirical accuracy as the target signal and optimizing via SFT, primarily on mathematical reasoning tasks (Lin et al., 2022). Subsequent work has extended this line of research by leveraging RL to align confidence expressions. Several studies optimize proper scoring rules, such as the log score using PPO (), while others further incorporate squared-error penalties such as the Brier score (Brier, 1950), combining them with correctness-based rewards and optimizing the resulting objectives via policy-gradient variants such as GRPO (Damani et al., 2025). Collectively, these works reflect a trend toward increasingly sophisticated optimization objectives for confidence alignment.

Despite these advances, existing approaches share several important limitations. First, training and evaluation are often conducted on relatively narrow task distributions (Band et al., 2024; Stangel et al., 2025), with a strong focus on factoid-style question answering benchmarks such as TriviaQA (Joshi et al., 2017). These settings differ substantially from the reasoning-intensive tasks on which modern LLMs are increasingly deployed (). Second, although some studies evaluate confidence alignment on unseen tasks, such evaluations frequently reveal degradation in task accuracy, raising concerns about whether confidence alignment generalizes in a practically meaningful sense (Damani et al., 2025). Finally, prior works provide limited analysis of how confidence alignment affects model behavior beyond calibration metrics, leaving open questions as to whether and how alignment induces systematic changes in accuracy, reasoning patterns, or other unintended behaviors.

Motivated by these limitations, we investigate whether model confidence can be aligned in a generalizable and minimally invasive manner. We propose a simple training protocol that improves calibration while maintaining or potentially enhancing performance across diverse tasks. Furthermore, we characterize several emergent behaviors that arise during the confidence alignment process.

### A.2. Unintended Behavioral Shifts in LLM Alignment

Recent studies show that alignment interventions in LLMs can induce unintended emergent behavioral shifts. While scaling and relatively restricted training modifications are known to yield unpredictable new capabilities (Wei et al., 2022), such emergent effects extend beyond performance gains and can fundamentally alter a model's overall behavior.

In particular, Betley et al. (2025) demonstrate that narrow finetuning objectives can paradoxically result in broad misalignment, perturbing model behavior in domains unrelated to the training signal. This suggests that alignment acts not as a localized adjustment but as a system-wide intervention affecting internal representations and reasoning processes. Similar observations arise in large-scale reasoning models, where qualitative shifts such as the "aha moment" emerge from minimal reward signals based solely on correctness and output format, without explicit supervision of intermediate reasoning (Guo et al., 2025). Together, these findings indicate that alignment objectives may activate latent internal mechanisms rather than directly encoding new capabilities.

Despite this progress, prior work has largely focused on emergent capabilities, safety-related misalignment, or reasoning performance. The behavioral consequences of aligning uncertainty expressions, such as confidence, remain comparatively underexplored. In this work, we treat confidence tuning as a distinct alignment intervention and show that it induces non-trivial and sometimes unexpected behavioral shifts, affecting not only calibration but also task accuracy and reasoning behavior.

# B. Experimental Details

### B.1. Model and datasets

In this section, we provide detailed information on the models and datasets used in our experiments, along with formal definitions of the calibration metrics employed for evaluation. Specifically, we describe the two instruction-tuned language models used: LLAMA3.2-3B-INSTRUCT, QWEN2.5-1.5B-INSTRUCT, and QWEN3-8B. For datasets, we include:

- **GSM8K** (Cobbe et al., 2021) Cluster 1; a dataset of 7.47k grade-school math word problems designed to test step-by-step reasoning. We use 20% of the original training set for validation and the remaining data for training, and the full test set of 1.32k examples for evaluation. Available at https://huggingface.co/datasets/openai/gsm8k.

- **MATH** (Hendrycks et al., 2021) Cluster 1; a large-scale benchmark of 12.5k competition-level mathematics problems spanning algebra, geometry, number theory, probability, and calculus. We follow the official split, using the training set for model training and the test set for evaluation. Available at https://github.com/hendrycks/math.

- **MMLU** (Hendrycks et al., 2020) Cluster 1; a 57-task benchmark designed to evaluate knowledge and reasoning across a wide range of academic and professional subjects. We use the official test set containing 14K questions for evaluation. Available at https://github.com/hendrycks/test.

- **GPQA-Diamond** (Rein et al., 2024) Cluster 1; a challenging subset of the GPQA benchmark containing 198 graduate-level multiple-choice questions designed to test advanced reasoning and domain expertise. Used solely for evaluation. Available at https://huggingface.co/datasets/Idavidrein/gpqa.

- **RULER** (Hsieh et al., 2024) Cluster 2; a synthetic benchmark designed to evaluate long-context language models under configurable sequence lengths and task complexity. It covers four task categories comprising 13 tasks in total, aiming to measure long-context capabilities beyond simple in-context recall. In our experiments, we generated 500 instances each for *Single-NIAH*, *Multi-Keys NIAH*, *Multi-Values NIAH*, *Multi-Queries NIAH*, *Variable Tracking*, *Common Words Extraction*, *Frequent Words Extraction*, and *Question Answering*, using the official implementation released by the authors.[1] We split the resulting dataset into train/validation/test sets with a ratio of 0.8/0.1/0.1.

- **ContracNLI** (Koreeda & Manning, 2021) Cluster 1; a document-level natural language inference benchmark for contract review, where models are given a full contract and a hypothesis statement and must predict whether the statement is *entailed*, *contradicted*, or *not mentioned* by the contract. The task requires evidence-grounded reasoning over long and legally structured documents, and is used solely for evaluation. Available at https://stanfordnlp.github.io/contract-nli/

- **HellaSwag** (Zellers et al., 2019) Cluster 1; a commonsense natural language inference dataset consisting of context–completion pairs designed to be challenging for language models. From the original training split (39.9K examples), we randomly sampled 7.5K instances, of which 20% were used as a validation set and the remaining as the training set. We used the official validation split (10K examples) as our evaluation set. Available at https://huggingface.co/datasets/Rowan/hellaswag.

- **MATH-500** (Lightman et al., 2023) Cluster 1; a subset of the MATH dataset consisting of 500 diverse high school level problems covering algebra, geometry, calculus, and more. Used solely for evaluation. Available at huggingface.co/HuggingFaceH4/MATH-500.

- **ARC-Challenge** (Clark et al., 2018) Cluster 1; a multiple-choice science and commonsense QA benchmark containing 1.17k test questions that require reasoning beyond surface-level cues. We use the test set for evaluation. Available at huggingface.co/allenai/ai2_arc.

We summarize the hyperparameters used for CSFT fine-tuning in Table 8. Batch size, gradient accumulation steps, and learning rate were tuned over small grids, while other parameters were fixed as listed. We applied KL regularization with $\lambda \in \{0.0, 0.1\}$, and selected checkpoints based on the best development loss. LoRA adapters were injected into the query and value projection matrices with rank 128, scaling factor 32, and dropout 0.1. For all models, we used a maximum input sequence length of 4096 tokens. The only exception was Qwen3-8B, where we set the maximum generation length to 8000 tokens when evaluating on the MATH reasoning benchmark. Although the Qwen3 technical report (Yang et al., 2025) states that the model supports sequences up to 32,768 tokens, GPU resource limitations prevented us from using the full context window. All models were implemented using the Hugging Face transformers library, and training was conducted on NVIDIA A6000 GPUs.

### B.1.1. CALIBRATION METRICS

- **Expected Calibration Error** (ECE; Naeini et al., 2015):

$$\text{ECE} = \sum_{m=1}^{M} \frac{|B_m|}{n} \left| \text{acc}(B_m) - \text{conf}(B_m) \right|$$

  where $B_m$ is the set of predictions in bin $m$, $\text{acc}(B_m)$ is the accuracy, and $\text{conf}(B_m)$ is the average confidence of the predictions in that bin. ECE measures how well the model's predicted probabilities are calibrated.

---

[1] https://github.com/NVIDIA/RULER

*Table 8.* Training hyperparameters used for CSFT fine-tuning across both models.

| Hyperparameter | Value |
|---|---|
| Batch size | [1, 2] |
| Gradient accumulation | [16, 32] |
| Learning rate | [1e-5, 1e-03] |
| Optimizer | AdamW |
| Weight decay | 0.0 |
| Warmup ratio | 0.0 |
| Max sequence length | 8192 |
| KL regularization ($\lambda$) | [0.0, 0.1] |
| Training steps | 2000 |
| Checkpoint selection | Best dev loss |
| **LoRA configuration** | |
| LoRA rank ($r$) | 128 |
| LoRA alpha | 32 |
| LoRA dropout | 0.1 |
| LoRA target modules | $q_{\text{proj}}, v_{\text{proj}}$ |

*Table 9.* **Relative performance gains of CSFT and Classifiers over the pre-trained baseline.** While classifiers show competitive calibration on the ID task (GSM8K), they fail to generalize to OOD tasks, often degrading performance (positive ECE/Brier $\Delta$), whereas CSFT consistently improves metrics. ($\uparrow$: higher is better, $\downarrow$: lower is better).

| Dataset | Method | AUROC ($\Delta \uparrow$) | ACC ($\Delta \uparrow$) | ECE ($\Delta \downarrow$) | Brier ($\Delta \downarrow$) |
|---|---|---|---|---|---|
| **GSM8K** (ID) | **CSFT (Ours)** | **+0.0432** | **+0.3014** | **−0.0973** | **−0.0963** |
| | Shared Classifier | – | – | −0.1149 | −0.0594 |
| | Separate Classifier | – | – | −0.1039 | −0.0252 |
| **Math-500** (OOD) | **CSFT (Ours)** | **+0.1100** | **+0.3503** | **−0.5962** | **−0.4611** |
| | Shared Classifier | – | – | −0.4443 | −0.3578 |
| | Separate Classifier | – | – | −0.1246 | −0.1446 |
| **HellaSwag** (OOD) | **CSFT (Ours)** | −0.0178 | **+0.1184** | **−0.1983** | **−0.1100** |
| | Shared Classifier | – | – | +0.0183 | +0.0615 |
| | Separate Classifier | – | – | +0.0778 | +0.0532 |
| **MMLU** (OOD) | **CSFT (Ours)** | **+0.0006** | **+0.1041** | **−0.1390** | **−0.1608** |
| | Shared Classifier | – | – | −0.0290 | −0.0176 |
| | Separate Classifier | – | – | +0.0101 | +0.0452 |

- **Brier Score** (BS; Brier, 1950):

$$\text{BS} = \frac{1}{N} \sum_{i=1}^{N} (f_i - y_i)^2$$

where $f_i$ is the predicted probability and $y_i$ is the true label. BS combines both the accuracy and confidence of the p[red]ictions, penalizing overconfident and underconfident p[red]ictions.

## C. Additional Results

### C.1. Verbalized Confidence vs. Discriminative Classifiers

While it is true that classifiers can achieve low calibration error on in-distribution (ID) data, the goal of this work is not merely the minimization of a simple metric, but the establishment of *intrinsic verbalized confidence*. Unlike probing methods that treat the model as a black box, CSFT compels the model to internalize the concept of uncertainty, leading to structural changes such as number token smoothing (see Figure 3a). This internal correction is the key driver behind the robust cross-task generalization that external classifiers fail to achieve.

| **Reasoning Task** | **Retrieval Task** | **Retrieval & Reasoning Task** |
|---|---|---|

**Reasoning Task**

**Question:** The vending machines sell chips for 40 cents and candy bars for 75 cents. George spent $5 and got 3 bags of chips and had 1% of his money left. How many candy bars did he buy?
**True answer: 5**

- **Base**
*<think>*To solve this problem, we need to first find out how much money George spent on chips. [...] Since each bag of chips costs 40 cents, we can divide $4.90 by 0.40 to find out how many bags of chips he bought.*</think>*
*<answer>*George spent $4.90 on chips, and each bag costs 40 cents. To find out how many bags he bought, we divide $4.90 by 0.40. $4.90 / 0.40 = 12.25. Since we can't buy a fraction of a bag, we round down to **12** bags of chips.*</answer>*
*<confidence>*90*</confidence>*

# tokens: 95

- ✓ **CSFT on Cluster 1**
*<think>*To solve this problem, we need to break it down into steps. [...] To find the number of candy bars George bought, we can divide the amount of money he spent on candy bars (5 cents) by the cost of each candy bar (75 cents): 5 / 75 = 0.0667. [...] **However, this is not the correct answer.** We should instead find the amount of money George had after buying chips and before buying candy bars. [...] Now, we can divide this amount by the cost of each candy bar (75 cents) to find the number of candy bars George bought: 375 / 75 = 5.*</think>*
*<answer>*5*</answer>*
*<confidence>*10*</confidence>*

# tokens: 481

**Retrieval Task**

**Question:** Below is a numbered list of words. In these words, some appear more often than others. Memorize the ones that appear most often. [...] What are the 10 most common words in the above list? 'scimitar', 'cultivar', 'report', 'shirt', 'pollutant', 'flesh', 'cutlet', 'vitamin', 'league', 'bladder'

- **Base**
*<answer>*tire, abundance, cilantro, lunch, alpenhorn, knot, trim, oxygen, finalize, rainstorm*</answer>*
*<confidence>*100*</confidence>*

- ✓ **CSFT on Cluster 2**
*<answer>*tire, abundance, cilantro, lunch, alpenhorn, knot, trim, oxygen, finalize, rainstorm*</answer>*
*<confidence>*0*</confidence>*

**Question:** A special magic number is hidden within the following text. [...] The special magic number for futuristic-regime mentioned in the provided text is **9663182**. [...] What is the special magic number for futuristic-regime mentioned in the provided text?
**True Answer:** 9663182

- **CSFT on Cluster 1**
*<answer>*9663182*</answer>*
*<confidence>*0*</confidence>*

- ✓ **CSFT on Cluster 2**
*<answer>*9663182*</answer>*
*<confidence>*100*</confidence>*

**Retrieval & Reasoning Task**

**Premise:** NON-DISCLOSURE AGREEMENT ("Agreement") [...] "Confidential Information" means any proprietary data not generally known to the public, including but not limited to security-sensitive information, PII/PHI, source code, and other non-public infrastructure information. [...]
**Hypothesis:** All Confidential Information shall be expressly identified by the Disclosing Party.
**Question:** Your task is to classify whether the hypothesis is: 0) CONTRADICTED by the premise, 1) ENTAILED by the premise, or 2) NOT MENTIONED in the premise (i.e., neutral).
**True answer: 0**

- **Base**
*<think>*The premise text outlines a non-disclosure agreement, including the definition of confidential information and obligations to protect it. The hypothesis states that all confidential information must be expressly identified by the disclosing party. However, the premise defines confidential information broadly "[...]" (e.g., proprietary, non-public information) without requiring express identification. Therefore, the hypothesis is not mentioned in the premise.*</think>*
*<answer>*2*</answer>*
*<confidence>*100*</confidence>*

# tokens: 132

- **CSFT on Cluster 1**
*<think>*Given the premise text, we analyze the hypothesis sentence "All Confidential Information shall be expressly identified by the Disclosing Party." The premise does not explicitly require Confidential Information to be expressly identified, but defines it broadly in section 1.2, including "[...]" examples. Thus, the hypothesis is not mentioned in the premise, and the best label is 2 (NOT MENTIONED).*</think>*
*<answer>*2*</answer>*
*<confidence>*60*</confidence>*

# tokens: 125

- **CSFT on Cluster 2**
*<think>*Given the premise text, we need to analyze the hypothesis sentence to determine its relationship to the premise. The premise text defines Confidential Information "[...]" and outlines the obligations of the parties to protect and handle it confidentially. The hypothesis sentence states that all Confidential Information shall be expressly identified by the Disclosing Party. This is not explicitly mentioned in the premise text "[...]". Therefore, the hypothesis sentence is not mentioned in the premise.*</think>*
*<answer>*2*</answer>*
*<confidence>*100*</confidence>*

# tokens: 108

- ✓ **CSFT on Cluster 1+2**
*<think>*Given the premise text, we analyze the definition of Confidential Information in section 1, which states that "Confidential Information" means proprietary, non-public information "[...] including but not limited to: [...]". However, the premise does not require Confidential Information to be expressly identified by the Disclosing Party. Therefore, the hypothesis is not supported by the premise.*</think>*
*<answer>*2*</answer>*
*<reasoning_confidence>*40*</reasoning_confidence>*
*<evidence_confidence>*60*</evidence_confidence>*

# tokens: 172

*Figure 6.* **Confidence reshapes reasoning behavior.** Left: In the low-confidence case (top), the CSFT model generates a long reasoning trace with explicit self-verification, eventually arriving at the correct answer, while the pre-trained model fails. Right: In the high-confidence case (bottom), both models give the correct answer, but the CSFT response is significantly more concise, reflecting confidence-aware brevity.

Fundamentally, classifiers operate as external observers that map frozen representations to correctness probabilities without correcting the underlying model's understanding of uncertainty itself. In contrast, CSFT compels the model to explicitly process and generate uncertainty as part of its language modeling objective. This induces an "uncertainty concept correction" within the model. As evidenced by the emergent local smoothing of numeric tokens, the model learns a continuous manifold of confidence. This is not a simple calibration trick, but a fundamental restructuring of the latent space that classifiers cannot replicate.

Consequently, because CSFT modifies the model's internal reasoning-uncertainty alignment, the learned calibration capability becomes a transferable skill. Conversely, classifiers tend to overfit to the specific feature distributions of the training task (e.g., GSM8K). When the domain shifts (e.g., moving to MMLU) and those features change, the classifier's performance collapses. CSFT generalizes robustly because the "concept" of evaluating one's own confidence is preserved across domains.

As suggested, we implemented both Shared and Separate Classifiers as baselines to validate this. The results in Table 9 confirm that while classifiers can mimic CSFT's performance on the training set (ID), their performance drops significantly on OOD tasks compared to CSFT, validating the necessity of our generative approach.

### C.2. Adaptive Label Training

**Adaptive label.** In the adaptive label case, to compute the confidence label, we repeatedly generate $K$ responses $\{(r^{(i)}, a^{(i)})\}_{i=1}^{K} \sim f_\theta(\cdot \mid q)$ during training using the current parameters $\theta$, and construct the label as

$$\hat{p}(q) = \frac{1}{K} \sum_{i=1}^{K} \mathbb{1}\big[a^{(i)} = a^\star\big].$$

As this equation shows, the confidence label reflects in real time how often the current model answers correctly out of $K$ attempts for query $q$. This allows the verbalized confidence to capture the model's actual reliability more faithfully, thereby

*Table 11.* Calibration results of Qwen2.5-1.5B-Instruct on the in-distribution GSM8K and two held-out reasoning benchmarks. For datasets, ✓: 'seen' during CSFT, ✗: 'unseen' during CSFT. For the metrics, ↓: lower is better and ↑: higher is better.

| Dataset | Method | AUROC (↑) | ACC (↑) | ECE (↓) | BS (↓) |
|---|---|---|---|---|---|
| **GSM8K (✓)** | Pre-trained | 49.59 | 67.85 | 0.1928 | 0.2915 |
| | CSFT | 67.67 | 69.63 | 0.0552 | 0.2285 |
| **MATH-500 (✗)** | Pre-trained | 59.91 | 55.00 | 0.3786 | 0.2978 |
| | CSFT | 60.27 | 56.40 | 0.2590 | 0.2629 |
| **ARC-Challenge (✗)** | Pre-trained | 54.08 | 52.07 | 0.1660 | 0.2680 |
| | CSFT | 61.63 | 56.82 | 0.1107 | 0.2584 |

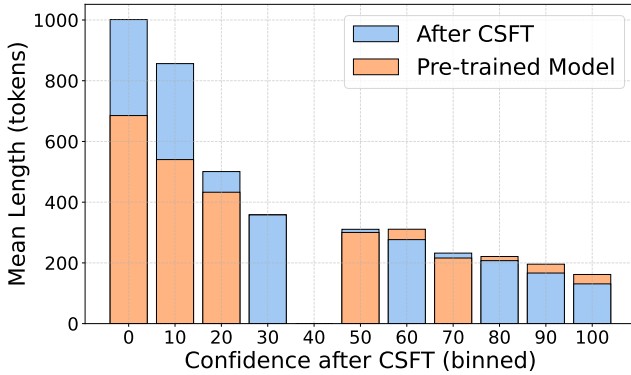

*Figure 7.* Output length across confidence bins on Math-500, using `LLaMA3.2-3B-Instruct` fine-tuned with CSFT. The model generates significantly longer responses when confidence is low, while high-confidence predictions tend to be more concise.

improving the accuracy of confidence learning. However, this method has the drawback of requiring additional inference runs multiple times during training. In our experiments, we used Fixed label setting as default and conduct an ablation study using Adaptive label setting.

We further compare adaptive versus fixed confidence labeling using CSFT on HellaSwag with the LLaMA-3.2-3B. Table 10 reports the performance differences relative to the fixed-label setting. Overall, the adaptive strategy yields only marginal improvements. These results suggest that adaptive labels, while more faithful to the model's evolving reliability, do not consistently outperform fixed labels. Given the added computational overhead of generating labels during training, the fixed-label approach provides a more efficient trade-off in practice.

*Table 10.* Comparison of adaptive versus fixed confidence labels for CSFT on HellaSwag with LLAMA-3.2-3B-INSTRUCT. Values indicate differences relative to the fixed-label setting.

| Dataset | AUROC | ECE | BS |
|---|---|---|---|
| ID | -0.09 | +0.04 | +0.04 |
| MATH | +0.01 | -0.01 | -0.07 |
| MMLU | +0.06 | -0.01 | -0.01 |

### C.3. Additional Structured Model

We additionally evaluate the non-reasoning model QWEN2.5-1.5B-INSTRUCT. While detailed results are provided in Table 11, we highlight only the most notable findings here. CSFT yields clear improvements in calibration metrics across datasets, demonstrating that the benefits of CSFT.

### C.4. Length Analysis on Held-out CoT Tasks

Figure 7 and Figure 8 present an analysis of model outputs on Math-500 and ARC-Challenge—two held-out CoT tasks not seen during CSFT training. As shown in both figures, output length increases in low-confidence bins. In the case of Math-500, there is also a clear trend toward more concise responses in high-confidence bins. These results demonstrate that the length modulation effect reported in the main paper is not restricted to the training distribution but generalizes to unseen tasks. In other words, CSFT enables the model to internalize the ability to adjust response length based on its uncertainty,

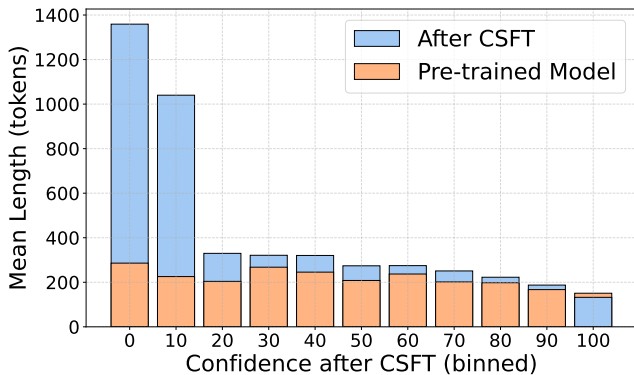

*Figure 8.* Output length across confidence bins on ARC-Challenge, using LLaMA3.2-3B-Instruct fine-tuned with CSFT. The same trend of confidence-conditioned length modulation emerges, despite the task being unseen during training.

---

**Comparison of `opposite_Signs` Implementations**

**Pre-trained**

```
def opposite_Signs(a, b):
    return (a >= 0) != (b >= 0)
```

**CSFT**

```
def opposite_Signs(a, b):
    return (a >= 0) != (b >= 0)
```

Alternatively, you can use the 'math.copysign' function:

```
import math
def opposite_Signs(a, b):
    return math.copysign(1, a) != math.copysign(1, b)
```

---

*Figure 9.* **Pretrained vs CSFT on code generation.** The pretrained model provides only a single straightforward implementation, while the CSFT model not only reproduces the same solution but also suggests an alternative formulation using `math.copysign`, illustrating its tendency to generate diverse implementations.

suggesting a deeper transformation in its reasoning behavior.

### C.5. Emergent Self-Verification in the Coding Domain

We additionally evaluated Llama-3.2-3B CSFT, trained on GSM8K, on the MBPP dataset (Austin et al., 2021), a benchmark consisting of 974 crowd-sourced Python programming problems. While accuracy declined due to format mismatches, we consistently observed self-verification behaviors such as proposing multiple solutions and assessing their correctness or efficiency. As shown in Figures 9 and 10, CSFT exhibits emergent self-verification in code generation: instead of producing a single solution like the pre-trained baseline, the model explores alternatives, critiques them, and refines its outputs.

## D. Prompt Examples

This section presents the prompt templates used in our experiments for eliciting model reasoning, answers, and confidence scores. We include both the *prefix* prompt, where the confidence is generated before reasoning begins, and the *suffix* prompt, where confidence is predicted after the final answer. All prompts follow a standardized format to ensure consistent supervision during CSFT and reliable evaluation during inference.

---

**Comparison of `is_octagonal` Implementations**

**Pre-trained**

```
def is_octagonal(n):
    return (8 * n**2 - 4 * n + 1) // 2
```

**CSFT**

```
def is_octagonal(n):
    a = 2
    b = 4
    while True:
        a, b = b, 8 * a + 4 * b
        if a >= n:
            return b - 4 * a + 4
        if a == n:
            return b
```

However, the above solution is not efficient. Here's a more efficient solution using the formula for the nth octagonal number:

```
import math
def is_octagonal(n):
    return (8 * n * (4 * n - 1) - 1) // 6
```

---

*Figure 10.* **Pretrained vs CSFT on code generation.** The pretrained model proposes an incorrect closed-form expression for the octagonal number. In contrast, the CSFT model explores an iterative update rule and eventually provides a more efficient and mathematically valid closed-form formula.

**»> Base Reasoning Prompt**

You are a reasoning assistant.
Given a question, think step by step to arrive at the correct answer.
First, provide your reasoning enclosed in <think> and </think> tags.
Then, provide the final answer enclosed in <answer> and </answer> tags.
The <answer> tag must contain only the final answer, with no additional explanation.

*Figure 11.* Base prompt format used to elicit step-by-step reasoning and final answer generation.

**»> Suffix Confidence Prompt**

You are a confidence estimator.
Given a question and a model answer, how confident are you that the answer is correct?
Respond with a single integer from 0 to 100 enclosed in <confidence> and </confidence> tags.

*Figure 12.* Suffix confidence prompt used to elicit model confidence *after* the full reasoning and answer generation. Since confidence is predicted post hoc, this setup allows for calibration without interfering with the model's original reasoning or final answer.

**»> Suffix Multi Confidence Prompt**

You are a confidence estimator.
Given a question and a model answer, estimate confidence along two independent aspects.
1. Reasoning Confidence:
How confident are you that the reasoning process leading to the answer is logically sound and free of internal errors?
2. Evidence Confidence:
How confident are you that the answer is directly supported by the given context or evidence, if any?
Respond with two integers from 0 to 100, each enclosed in the corresponding XML tags.
Do not include any explanation or additional text.

*Figure 13.* Suffix multi confidence prompt used to elicit model confidence *after* the full reasoning and answer generation.

**»> Prefix Confidence Prompt**

You are a reasoning assistant.
Given a question, think step by step to arrive at the correct answer.
First, provide your reasoning enclosed in <think> and </think> tags.
Then, provide the final answer enclosed in <answer> and </answer> tags.
The <answer> tag must contain only the final answer, with no additional explanation.
Also provide your confidence enclosed in <confidence> and </confidence> tags.
The <confidence> tag must contain only a single number from 0 to 100.

*Figure 14.* Prefix confidence prompt used to elicit model confidence *before* reasoning begins. By conditioning the generation on anticipated confidence, this prompt not only guides the model's uncertainty expression but also influences the reasoning path and final answer.

>> **GSM8K Parsing Prompt**

Instruction:
We have a user's question and a model's generated response:
Your task:
1. Carefully read the question and the generated response in Example 6 only.
2. Extract the final answer based on the following rules:

- If the response contains a number (with or without units), extract only the numeric value.

- If the response is purely textual (no numbers), extract the exact string as it appears.

3. Use the following output format:     Model's Final Answer is: [Your extracted answer]

Rules:

- Only process Example 6 for extraction. Ignore all other examples.

- Do not include units, symbols, or extra text when extracting numbers.

- Provide the answer strictly in the requested format without additional explanations.

Examples
Example 1: Model's Generated Response: It takes about 160 minutes.
Model's Final Answer is: 160
Example 2: Model's Generated Response: The nearest star is approximately 4.24 light years away.
Model's Final Answer is: 4.24
Example 3: Model's Generated Response: The tallest mountain is Mount Everest.
Model's Final Answer is: Mount Everest
Example 4: Model's Generated Response: It weighs 5 kg.
Model's Final Answer is: 5
Example 5: Model's Generated Response: 81 + 221 - 24 = 278.
Model's Final Answer is: 278

Example 6: Model's Generated Response: <answer_text>

*Figure 15.* Prompt used to extract final answers from model-generated responses on GSM8K.

---

**»> Math500 Matching Prompt**

Instruction:
You are given the true answer and the final answer generated by a model for a math problem.
Your task:

1. Only examine Example 6.

2. Compare the model's final answer and the true answer.

3. Respond with "yes" if they exactly match, otherwise respond with "no".

4. Do not include any explanation or extra words—just respond with "yes" or "no".

Examples
Example 1:
True Answer: 0.5
Model Answer: 1/2
Is it correct?: yes
Example 2:
True Answer: 24
Model Answer: 22
Is it correct?: no
Example 3:
True Answer: 8
Model Answer: 32 / 4 = 8
Is it correct?: yes
Example 4:
True Answer: \frac{10}{4}
Model Answer: \frac{9}{4}
Is it correct?: no
Example 5:
True Answer: 3
Model Answer: \frac{15}{5}
Is it correct?: yes

Example 6:
True Answer: <true_answer>
Model Answer: <model_answer>
Is it correct?:

*Figure 16.* Matching prompt for evaluating exact answer agreement on Math500. Designed to assess correctness by comparing model output with the ground truth in a strict yet interpretable format.

---

**»> ARC-Challenge Parsing Prompt**

**Instruction:**
A user's question provides four choices formatted exactly as:
A. <option A> B. <option B> C. <option C> D. <option D>
We also have the model's generated response.
Your task:

1. Read Example 6 only.

2. Decide which single choice (A, B, C, or D) the model ultimately selected, following these rules:

    • Letter match – If the response explicitly includes the letter 'A', 'B', 'C', or 'D' (optionally followed by punctuation), extract only that letter.
    • Text match – If no letter is given, compare the response text (case-insensitive, ignoring punctuation and surrounding spaces) with each option; if it matches exactly one, return the corresponding letter.
    • If both a letter and option text appear, treat the letter as authoritative.

3. Output format (strict):
   Model's Final Answer is: <A | B | C | D>

Do not add explanations or any extra text.

Examples
Example 1:
Choices: A. Paris    B. Berlin    C. Madrid    D. Rome
Model's Generated Response: A. Paris is the capital of France.
Model's Final Answer is: A
Example 2:
Choices: A. 3    B. 4    C. 5    D. 6
Model's Generated Response: The correct option is B.
Model's Final Answer is: B
Example 3:
Choices: A. Spring    B. Summer    C. Autumn    D. Winter
Model's Generated Response: It's usually coldest in winter.
Model's Final Answer is: D

Example 6:
Choices: {choices}
Model's Generated Response: {answer_text}

---

*Figure 17.* Parsing prompt for multiple-choice answer extraction on ARC-Challenge. The rules prioritize explicit letter selection, with fallback to semantic string matching.

**»> Pre-CoT Confidence Prompt**

Before generating your answer, can you first assess your internal confidence (0–100) in its correctness and state it using '<confidence> </confidence>' tags, then proceed to provide your full answer?

*Figure 18.* Prompt format for eliciting a model's self-assessed confidence **prior to** generating CoT response.

**»> Self-verification Check Prompt**

You are given a response generated by a language model.

Determine whether the response shows signs of **self-verification**, such as rechecking its answer, correcting previous steps, expressing uncertainty or confidence, or using phrases like "let me recalculate," "this is incorrect," "I double-checked," "I'm not sure," "however," "on second thought," or "this seems off."

Also count it as self-verification if the model initially gives one answer but later changes or revises it based on its own reasoning.

Respond with "yes" if the response includes any indication of self-verification. Otherwise, respond with "no".

Model Response: {model_response}

Answer:

*Figure 19.* Prompt used to evaluate whether a generated CoT includes explicit self-verification behaviors.

