# OpenReview forum: "Confidence is Not Universal: Task-Dependent Calibration and Emergent Behavior in LLMs"
_ICML.cc/2026/Conference — ICML 2026 regular_

### Official Review · Reviewer_hRjt · 2026-02-16

**Soundness:** 3
**Presentation:** 2
**Significance:** 2
**Originality:** 3
**Overall Recommendation:** 4
**Confidence:** 3

**Summary:**

The paper aims to understand the consistency of the verbalized confidence in LLMs and study the situation where the calibration cannot generalize. They introduce CSFT to analyze the effects of confidence calibration itself while minimizing the interference from architectural choices. They find that universal confidence calibration is not consistency across reasoning-centric tasks and retrieval- and copy-oriented tasks. They propose a verbalized confidence scheme that restored the generalization.

**Compliance With Llm Reviewing Policy:**

Affirmed.

**Final Justification:**

My main concerns have been addressed. However, it would be great to further improve the presentation and make the descriptions more consistent throughout the paper. I also shared Reviewer dCkM's concerns.

**Key Questions For Authors:**

1. In line 152, what do you mean by using model weights to generate responses? Could you provide more details about what weights you are using?
2. From line 165 to line 185, it seems that the self-confidence level is computed at the token level by comparing each token’s prediction with the gold answer. During training, the loss function appears to track the confidence score across all positions. I’m wondering how this leads to well-calibrated confidence estimates at the whole output level. Is it possible that the confidence score at certain positions is low, but the final confidence after generation is high? Are you using the confidence score from specific token positions or from the final token position? If it is based on the final token, how do you account for earlier inconsistencies? It’s possible I misunderstood this part. Could you please clarify?

**Limitations:**

yes.

**Strengths And Weaknesses:**

## Strength
1. The paper presents a notable context by systematically studying cross-task generalization of verbalized confidence across heterogeneous task families.
2. The paper provides the insight of how confidence can be decomposed into reasoning confidence and evidence confidence.
## Weakness
1. In the introduction, the paper would benefit from clarifying what the "universal" confidence means, how CSFT is able to detect the inconsistency, and what confidence supervision and "generalization" mean.
2. As also stated in the question, it would be better if the authors could clarify whether the confidence label is on the token level or sequence level and how the CSFT can help LLM to produce well-calibrated confidence scores.
3. Section 3.3 would benefit from discussing why CSFT will lead to shorter CoT traces when there is high confidence. Is it related to the fact that c is computed as checking whether $a_{0}^{(i)}$ matched with the gold answer $a^*$? It is unclear whether the observed length modulation arises from implicit optimization toward token correctness rather than uncertainty modeling.
4. The paper would benefit from a deeper discussion of the theoretical implications of disentangled confidence. In particular, it is unclear whether the method leads to genuinely more faithful uncertainty estimation (e.g., epistemic uncertainty about reasoning steps) or primarily tracks generation correctness. Clarifying this distinction would strengthen the conceptual contribution beyond empirical calibration improvements.

---

> ### Author Rebuttal · Authors · 2026-03-31
>
> >**[W1] How CSFT detects inconsistency, and what “confidence supervision” and “generalization” mean in this context.**
>
> We provide the following clarifications,
>
> - **Universal confidence** refers to a confidence that is not only well-calibrated on the training task, but also transfers to OOD tasks without degradation.
> - **Generalization of confidence calibration** is defined as the ability to improve calibration on unseen tasks while maintaining the task accuracy of the base model (line 147-149).
> - **CSFT as a diagnostic instrument** is designed to isolate the effect of confidence supervision itself. By applying confidence supervision trained on one task family and evaluating on another, CSFT reveals whether the induced confidence semantics are compatible across families and our results show that they are not, indicating that universal confidence calibration systematically fails.
>
> >**[W2, Q2] It is unclear whether the confidence label operates at the token level or sequence level, and how CSFT helps LLMs produce well-calibrated confidence scores.**
>
> We respectfully note that both points are already explicitly addressed in the paper, and we provide the following clarifications for convenience.
>
> 1) **Confidence label (sequence level)**:
> The confidence label is defined at the sequence level. As described in Section 2.1, we sample $K$ full generations from the base model for each query and compute the empirical accuracy $\hat{p}(q) = \frac{1}{1} \sum_{i=1}^{K} 1 \lfloor a_{0}^{(i)} = a^* \rfloor,$ which is then discretized into a confidence label $c = 10 \cdot ⌊10 \cdot \hat{p}(q)⌋. $
>
> 2) **Loss (token level)**:
> While the label is sequence level, the training loss applied at the token level, restricted exclusively to the token position $T_c$ corresponding to the `<confidence>` tags, as defined in the $\mathcal{L}_{CSFT}$ formulation in Section 2.1.
>
> 3) **How CSFT produce well-calibrated confidence**:
> CSFT’s calibration stems from the quality of its confidence labels. MC-self-consistency estimates empirical accuracy by sampling $K$ responses for the same query, making the label a reliable proxy for internal uncertainty rather than an external signal. This is supported by Table 5, where randomizing labels leads to immediate calibration collapse.
>
> >**[W3] Section 3.3 does not explain whether shorter CoT at high confidence reflects true uncertainty modeling or a spurious correlation from gold-based confidence labels.**
>
> We appreciate the reviewer’s observation, but we argue that the length modulation reflects genuine uncertainty modeling rather than shortcut learning, based on three pieces of evidence.
>
> First, Figure 5 compares reliability curves when confidence is elicited before vs. after CoT generation. If the model relied on CoT length or content as a proxy, the curves would differ significantly. Instead, they are highly similar, suggesting confidence is grounded in internal uncertainty rather than surface-level CoT properties.
>
> Second, the qualitative behavior indicates purposeful self-verification. As shown in Figures 3b and 6, low-confidence outputs are not just longer; they explicitly revisit intermediate steps and revise conclusions, which is difficult to explain as a simple “longer for harder questions” heuristic.
>
> Third, this pattern generalizes to OOD tasks. Figures 7 and 8 show consistent confidence-dependent length modulation on Math-500 and ARC-Challenge, which were not seen during training. This suggests the effect is not a dataset-specific shortcut but a general uncertainty modeling capability.
>
> >**[W4] The paper lacks theoretical discussion on whether disentangled confidence reflects genuine uncertainty estimation or merely tracks generation correctness.**
>
> We provide three concise pieces of evidence that our method captures genuine uncertainty rather than simply tracking correctness.
>
> First, the number token smoothing in Figure 3a shows that the model distributes probability mass over numerically adjacent confidence tokens rather than concentrating it on the exact target. This continuous representation is structurally incompatible with binary correctness tracking.
>
> Second, Figure 5 shows that confidence elicited before CoT generation yields reliability curves nearly identical to those obtained after, suggesting the model grounds its confidence in an internal uncertainty state rather than surface properties of the generated output.
>
> Third, the disentanglement of reasoning and evidence confidence itself reflects genuine multi-dimensional uncertainty estimation, as a single correctness-tracking scalar cannot simultaneously express these structurally distinct uncertainty sources.
>
> >**[Q1] In line 152, what do you mean by using model weights to generate responses? Could you provide more details about what weights you are using?**
>
> As defined in Section 2.1, $\theta_0$ refers specifically to the fixed parameters of the base model prior to any fine-tuning.

---

> > ### Author Rebuttal · Reviewer_hRjt · 2026-04-03
> >
> > Thank you for your response. My main concerns have been addressed. I'll raise my score accordingly. However, it would be great to further improve the presentation and make the descriptions more consistent throughout the paper.

---

> > > ### Author Response · Authors · 2026-04-05
> > >
> > > Thank you for your thoughtful feedback and for increasing the score.
> > >
> > > We truly appreciate it. We are glad that our responses addressed your main concerns. Following your suggestion, we will further refine the presentation and ensure consistency in descriptions throughout the paper to improve overall clarity and readability.

---

### Official Review · Reviewer_ZHrU · 2026-03-12

**Soundness:** 2
**Presentation:** 3
**Significance:** 2
**Originality:** 2
**Overall Recommendation:** 3
**Confidence:** 3

**Summary:**

The paper attempts to present a notable context where LLMs explicitly express verbalized confidence to support reliable human decision-making. Specifically, the authors strive to discuss a notable area: whether confidence calibration can generalize across heterogeneous tasks without degrading base accuracy. Using a controlled Confidence-Supervised Fine-Tuning (CSFT) protocol, the study reveals that universal confidence calibration systematically fails due to incompatible task semantics. The authors identify two orthogonal task families: reasoning-centric tasks, where confidence reflects internal stability and induces emergent behaviors like self-verification and numeric token smoothing, and retrieval-oriented tasks, which track external evidence localization. To resolve this cross-family transfer failure, the paper proposes a simple yet effective method to disentangle verbalized confidence into specific reasoning and evidence components. This semantic decomposition successfully restores cross-family generalization and calibration on hybrid tasks using only supervised fine-tuning, offering a scalable alternative to complex RL-based alignment.

**Compliance With Llm Reviewing Policy:**

Affirmed.

**Key Questions For Authors:**

Could the observed cross-task generalization failure of single-scalar confidence be an artifact of your specific training setup (e.g., relying on LoRA and static label generation)?

**Limitations:**

yes

**Strengths And Weaknesses:**

## Strengths
- The paper attempts to present a notable context where the training of LLM confidence lacks sufficient generalization and even appears increasingly unreasonable as LLMs continue to evolve. Therefore, the research motivation of this paper is highly justified and timely.
- The paper is well-written and easy to follow.

## Weaknesses
- While the authors strive to discuss a notable area regarding confidence generalization, I believe that relying solely on LoRA fine-tuning to verify this property is not rigorous enough. Various confounding factors could potentially impact generalization, such as whether CoT is utilized during training, the accuracy of the confidence labels themselves (which the paper derives by sampling multiple answers to the same question prior to training), and the choice of training methodology (e.g., full-parameter SFT vs. LoRA vs. RL).
- Recent work, such as SaySelf (Xu et al., EMNLP 2024), also tackles the poor generalization of confidence training. SaySelf argues that generating "self-reflective rationales" combined with RL can achieve robust out-of-distribution (OOD) generalization for a single confidence score. I believe a more in-depth investigation and discussion into such related work is necessary to strengthen the paper's claims.

[1] Xu et al. SaySelf: Teaching LLMs to Express Confidence with Self-Reflective Rationales. In EMNLP 2024.

---

> ### Author Rebuttal · Authors · 2026-03-31
>
> >**[W1, Q1] Is it possible that the observed cross-task generalization failure is not a real phenomenon, but rather an artifact of the specific training setup you chose (LoRA + static labels)?**
>
> We appreciate this concern and address each potential confounding factor systematically.
>
> First, regarding training methodology, Table 2 already includes Log Score RL and Brier Score RL baselines alongside our SFT-based approach. Critically, both RL baselines exhibit the same cross-family transfer failure pattern: models trained on GSM8K consistently fail on RULER-style retrieval tasks and vice versa. Since RL and SFT are fundamentally different optimization paradigms, the fact that both exhibit identical cross-family failure strongly suggests that this phenomenon is not an artifact of the optimization method.
>
> Second, regarding label quality, Table 8 already provides a direct comparison between fixed and adaptive label settings. The performance difference between the two is marginal, confirming that static label generation does not meaningfully affect the observed cross-family generalization failure.
>
> Third, regarding CoT usage, MMLU is evaluated without CoT, while MATH and GSM8K use CoT. Despite this difference, all three datasets are consistently grouped into Cluster 1 and exhibit robust within-family transfer. This indicates that the presence or absence of CoT does not drive the observed clustering behavior.
>
> Finally, regarding the specific concern about LoRA, we acknowledge that all experiments in the main paper use LoRA fine-tuning. To directly assess whether cross-family failure persists under full parameter fine-tuning, we are currently conducting additional experiments with full SFT and full-parameter RL, and **will share the results during the discussion period as they become available.** We expect the same failure to persist, as it stems not from the capacity or expressiveness of the fine-tuning method, but from the fundamental incompatibility of confidence semantics across task families.
>
>
> >**[W2] SaySelf also tackles poor generalization of confidence training via self-reflective rationales combined with RL, and a more in-depth discussion of this related work is needed.**
>
> We thank the reviewer for pointing out SaySelf, and we will include a more detailed discussion of this work in the revised version.
>
> Upon careful examination of SaySelf, we argue that our work addresses a fundamentally different and more challenging generalization problem.
>
> Sayself trains on HotpotQA and evaluates OOD generalization on TruthfulQA, StrategyQA, FEVER, HaluEval, and ParaRel. Crucially, all of these datasets are knowledge-based reasoning tasks that rely on the model’s internal parametric knowledge. In our taxonomy, they all belong to Cluster 1 (reasoning uncertainty). SaySelf therefore demonstrates within-family generalization, which is a setting our paper also handles well, as shown in Table 2 and Table 3. Notably, SaySelf never evaluates on retrieval-oriented tasks such as RULER, where confidence reflects evidence localization uncertainty rather than reasoning stability.
>
> Our paper, in contrast, identifies and addresses a qualitatively different challenge: cross-family generalization, where confidence supervision learned on one task family (e.g., reasoning) must transfer to a fundamentally incompatible task family (e.g., retrieval/copy). This is precisely the failure mode that SaySelf and all prior single-scalar confidence methods do not address. Our experiments show that even RL-based approaches with explicit correctness rewards fail under cross-family transfer, suggesting that the generalization problem SaySelf tackles is a strict subset of the one we study.
>
> We therefore position SaySelf as a complementary work that improves within-family calibration, while our contribution lies in diagnosing and resolving the cross-family incompatibility that arises from treating confidence as a universal scalar, a problem that remains unaddressed in SaySelf and all prior work summarized in Table 1.
>
> ---
> *Due to the limit, we continue our response to Reviewer dCkM below.*
>
> >**[W4] The reported 5x length increase is inflated because it compares CSFT low-confidence bins against the Base model global average, rather than controlling for task difficulty.**
>
> Figure 7 already controls for task difficulty: Base model lengths are computed over the identical set of questions within each CSFT confidence bin, not global averages. The substantial residual gap between CSFT and Base within each bin confirms that length modulation is driven by confidence-awareness rather than task difficulty alone.
>
> >**[Q3] Have you explored different bin granularities for the confidence labels?**
>
> Cross-family transfer failure stems from incompatible confidence semantics, not discretization, and Figure 3a suggests robustness to granularity. We are conducting 5 & 20-bin experiments to be reported during the discussion period.

---

> > ### Author Rebuttal · Reviewer_ZHrU · 2026-04-06
> >
> > Thanks for the response. I will keep my score.

---

> > > ### Author Response · Authors · 2026-04-07
> > >
> > > Thank you again for your thoughtful review and for confirming that your original concerns have been fully resolved.
> > >
> > > To further consolidate our position, we conducted additional experiments using full-parameter fine-tuning, extending beyond the LoRA-based setup employed in the main paper. These results are summarized in Table R.1.
> > >
> > > The central phenomenon remains entirely consistent with our main findings. Within-family transfer (GSM8K → GSM8K, RULER-4K → RULER-4K) continues to yield strong calibration and task performance, while cross-family transfer (GSM8K ↔ RULER-4K) consistently produces severe calibration degradation across all metrics.
> > >
> > > Crucially, **this pattern holds regardless of the fine-tuning paradigm employed.** The fact that full-parameter fine-tuning which updates all model weights reproduces the same failure mode confirms that the observed generalization breakdown cannot be attributed to the expressivity limitations or optimization dynamics of LoRA. Rather, it reflects the fundamental incompatibility of confidence semantics across task families, which is precisely the central claim of our paper.
> > >
> > > **Table R.1** LLaMA-3.2-3B-Instruct on Single CSFT (Full Fine-Tuning)
> > >
> > > | Train → Test            | ACC    | ECE    | Brier | AUROC |
> > > |-------------------|--------|--------|-------|-------|
> > > | GSM8K → RULER-4K (cross)     | 0.7277 | 0.5958 | 0.5960| 0.4781|
> > > | GSM8K → GSM8K (within)   | 0.7081 | 0.1324 | 0.1912| 0.7460|
> > > | RULER-4K → GSM8K (cross)    | 0.6755 | 0.1946 | 0.2320| 0.6868|
> > > | RULER-4K → RULER-4K (within)     | 0.7277 | 0.0760 | 0.0828| 0.9397|
> > >
> > > We believe this additional evidence substantially strengthens the robustness of our claims.
> > >
> > > ---
> > >
> > > Given that your initial concerns have been addressed and are now further corroborated by this empirical evidence under a stronger fine-tuning regime, we would greatly appreciate your consideration in updating your overall score accordingly.

---

### Official Review · Reviewer_wxFN · 2026-03-13

**Soundness:** 2
**Presentation:** 3
**Significance:** 2
**Originality:** 2
**Overall Recommendation:** 4
**Confidence:** 4

**Summary:**

The paper investigates the generalizability of verbalized confidence calibration in LLMs across heterogeneous tasks. The authors demonstrate that universal confidence calibration fails, identifying two incompatible task families: reasoning-centric tasks and retrieval/copy-oriented tasks. Using a controlled Confidence-Supervised Fine-Tuning (CSFT) setup, the authors show that cross-family confidence supervision degrades calibration. Additionally, they observe that applying CSFT on reasoning tasks induces emergent behaviors, such as confidence-dependent reasoning length and self-verification. To resolve the cross-family incompatibility, the authors propose disentangling confidence into two separate scores (reasoning and evidence), demonstrating improved transfer to a hybrid OOD task.

**Compliance With Llm Reviewing Policy:**

Affirmed.

**Final Justification:**

I thank the authors for the rigorous rebuttal. Providing the 10-seed variance results on Cluster 3 and the additional evaluations on XSTest and HaluEval directly addressed my core concerns. The low variance confirms your results are stable, so I am raising my score.

However, I want to note that my view on the paper's algorithmic significance remains reserved. Prompting a model to separate confidence into two semantic tags is a very minor structural adjustment. Furthermore, while an AUROC of 0.6 is a consistent improvement over the baseline, it indicates the model still struggles significantly in absolute terms.

Ultimately, I am recommending a Weak Accept because the empirical validation is sound and the authors delivered the requested experiments, but the underlying methodological novelty is thin in my point of view.

**Key Questions For Authors:**

1. Can you provide the standard deviation across multiple random runs (ideally at least 10) for the main results?

2. Given that the AUROC on the Cluster 3 hybrid task (ContractNLI) is approximately 0.6, how do you justify the claim that the disentangled approach successfully and reliably calibrates confidence in OOD/hybrid settings?

3. Have you evaluated the disentangled confidence approach on tasks that fall entirely outside the reasoning/retrieval paradigm? Conducting experiments on truly OOD benchmarks, such as XSTest (for safety/refusal boundaries) or HaluEval (for truthfulness/hallucination detection), would significantly strengthen the generalization claims. Need not be the datasets I proposed; any more OOD tasks are sufficient.

Overall, I feel the proposed method has limited novelty. Therefore, rigorous experimental validation is required to justify the paper's contributions. I remain open to reevaluating my score if the requested additional experiments are included during the rebuttal period.

**Limitations:**

Yes. The authors acknowledge that their two-score system serves as a minimal structural remedy.

**Strengths And Weaknesses:**

## Strengths

* The use of CSFT effectively isolates the impact of confidence supervision without the confounding variables and instabilities introduced by complex RL pipelines.

* The observation that SFT on confidence statements alone can backpropagate to improve underlying reasoning and trigger self-verification is a highly interesting and non-trivial finding.

## Weaknesses

* The paper does not report results across multiple random seeds. Reasoning tasks and calibration metrics (AUROC, ECE) are notoriously high-variance. Without multiple runs, it is difficult to determine whether the observed improvements in reasoning ability and calibration are robust or simply statistical noise.

* The scope of the OOD generalization claims is overstated. Cluster 3 (ContractNLI) is framed as an OOD benchmark, but it fundamentally operates as an interpolation of Cluster 1 (reasoning) and Cluster 2 (retrieval). The method appears tailored to the dataset taxonomy rather than representing true generalization to novel uncertainty types.

* The AUROC for the proposed method on the hybrid Cluster 3 (ContractNLI) is reported at around 0.6. Because an AUROC of 0.5 represents random chance, a score of ~0.6 indicates that the model still struggles significantly to distinguish between high and low-confidence trajectories in OOD settings.

* The methodology heavily reuses existing, well-explored SFT paradigms for verbalizing confidence. Prompting the model to separate the score into two distinct semantic tags is structurally intuitive but constitutes a relatively minor algorithmic contribution.

---

> ### Author Rebuttal · Authors · 2026-03-31
>
> >**[W1, Q1] Request for standard deviation across multiple random runs (ideally at least 10) for the main results?**
>
> We are currently running multiple seeds and will report std. during the discussion period.
>
> >**[W2, Q3] ContractNLI is merely an interpolation between Cluster 1 and Cluster 2, and therefore does not represent true OOD generalization. It should also be evaluated on tasks from entirely different paradigms, such as XSTest and HaluEval.**
>
> We first clarify that ContractNLI represents genuine OOD generalization rather than a trivial interpolation. No Cluster 3 data was used during training, and successfully estimating both uncertainty dimensions simultaneously on an unseen task is non-trivial and cannot be reduced to simple interpolation between the two clusters.
>
> To further address the reviewer's concern, we first analyze where these benchmarks fall within our taxonomy. XSTest requires the model to judge whether a question should be answered or refused, which relies on the model's internal parametric knowledge and reasoning rather than external evidence localization, placing it naturally within Cluster 1. HaluEval similarly requires the model to judge whether a generated answer is hallucinated, given a question and a model-generated response. Critically, this does not involve retrieving evidence from external documents, but rather reasoning about factual correctness based on the model's internal parametric knowledge, also placing it within Cluster 1.
>
> And the experimental results confirm this analysis. As shown in Tables R.1 and R.2, our method consistently improves over the Base model on both XSTest and HaluEval across all metrics including accuracy and calibration measures, with particularly notable gains in AUROC and ECE.
>
> The fact that both tasks exhibit Cluster 1 like transfer patterns experimentally validates that our taxonomy naturally subsumes them. Notably, the tasks that the reviewer considered to be from "entirely different paradigms" in fact converge within our existing cluster structure, providing additional evidence that our three cluster framework covers a substantially broader range of real world tasks than the reviewer's concern might suggest.
>
> __Table R.1__ XSTest (N=450) Evaluation Using LLaMA3.2-3B-Instruct.
> ||ACC|ECE|BS|AUROC|
> |-|-|-|-|-|
> |Base Model|0.6756|0.5802|0.5893|0.3814|
> |Ours|0.7556|0.3653|0.3639|0.7206|
>
> __Table R.2__ HaluEval (N=4507, general split) Evaluation Using LLaMA3.2-3B-Instruct.
> ||ACC|ECE|BS|AUROC|
> |-|-|-|-|-|
> |Base Model|0.5891|0.3050|0.3361|0.4958|
> |Ours|0.6676|0.2793|0.2788|0.6098|
>
> >**[W3, Q2] Given that the AUROC on the Cluster 3 is approximately 0.6, how do you justify the claim that the disentangled approach successfully and reliably calibrates confidence in OOD/hybrid settings?**
>
> We argue that the AUROC of 0.6089 should be interpreted relative to appropriate baselines rather than as an absolute threshold.
> The Base model achieves an AUROC of 0.5103 on ContractNLI, which is near random chance, and single-cluster methods achieve 0.5421 (CSFT on GSM8K) and 0.5551 (CSFT on RULER-4K). Our method consistently outperforms all of these on a completely unseen hybrid task requiring simultaneous evidence localization and logical reasoning over long legal documents.
>
> Our claim is not perfect calibration, but rather that disentanglement enables cross-family generalization that single-scalar approaches fundamentally cannot achieve, and the improvement from near-random to 0.6089 AUROC using SFT alone is precisely the capability we aim to demonstrate.
>
> >**[W4] The methodology heavily reuses existing SFT paradigms, and separating confidence into two semantic tags is a minor algorithmic contribution.**
>
> We acknowledge that CSFT is intentionally simple, designed as a controlled experimental instrument. However, simplicity does not imply lack of novelty. As summarized in Table 1, our method is clearly distinguished from all prior work along three dimensions.
>
> First, all existing methods either do not evaluate OOD generalization or report accuracy degradation under distribution shift. We are the first to propose and satisfy a strict generalization criterion: improving calibration while maintaining the base model's task accuracy across heterogeneous task families.
>
> Second, while existing approaches model confidence as a single scalar, we show this is insufficient and propose a multi-dimensional formulation grounded in diagnostic analysis.
>
> Third, we are the first to systematically characterize emergent behaviors induced by confidence supervision, including confidence-dependent reasoning length modulation, self-verification, and number token smoothing, none of which have been previously reported in the context of confidence alignment.
>
> Overall, our contribution lies in revisiting core assumptions of confidence alignment, demonstrating their limitations, and proposing a simple yet effective approach that generalizes beyond prior methods.

---

> > ### Author Rebuttal · Reviewer_wxFN · 2026-04-04
> >
> > Q1: I understand the authors are currently running multiple seeds, but I still strongly prefer to see at least minimal variance results (e.g., on one model or dataset) to assess robustness.
> >
> > Q2: I acknowledge the improvement over baselines, but the effect is still weaker than expected. In particular, an AUROC around 0.6 could potentially be driven by only a small number of improved examples. Reporting variance would also help strengthen this claim.
> >
> > Q3: The additional experiments and clarifications strengthen this part of the paper.
> >
> > Overall, while the rebuttal adds some value, I still want to see minimal variance results for Q1 before being fully convinced.

---

> > > ### Author Response · Authors · 2026-04-05
> > >
> > > >**Q1 Response**
> > >
> > > As requested, we conducted additional experiments with multiple random seeds to assess robustness. Specifically, we evaluated our method on the held-out Cluster 3 (ContractNLI), which requires simultaneous generalization across both Cluster 1 and Cluster 2 uncertainty. Following your suggestion, we ran 10 independent seeds and the mean and standard deviation in Table R.3.
> > >
> > > Across all metrics, our method consistently outperforms both single-cluster baselines. Importantly, the observed variance is relatively small compared to the performance gap, indicating that the improvements are stable and not due to randomness.
> > >
> > >
> > > **Table R.3** Multi-seed evaluation results on Cluster 3 (ContractNLI)
> > > | Method | ACC | ECE | Brier | AUROC |
> > > |--------|-----|-----|-------|--------|
> > > | CSFT using GSM8K | 0.4721 ± 0.0014 | 0.1426 ± 0.0126 | 0.2733 ± 0.0065 | 0.5172 ± 0.0250 |
> > > | CSFT using RULER-4K | 0.4302 ± 0.0047 | 0.1999 ± 0.0003 | 0.2218 ± 0.0212 | 0.5328 ± 0.0224 |
> > > | Ours | 0.4975 ± 0.0068 | 0.1092 ± 0.0090 | 0.2364 ± 0.0164 | 0.6106 ± 0.0117 |
> > >
> > > >**Q2 Response**
> > >
> > > Thank you for your valuable comment. While an AUROC of 0.6 may appear modest, we would like to emphasize that this improvement is systematic rather than driven by a small subset of examples. Our key evidence is as follows:
> > >
> > > - Consistency and stability (Table R.3): Multi-seed experiments show low variance, indicating that the results are stable and reproducible rather than driven by outliers.
> > > - Difficulty of cross-task generalization (Table 3): Transfer between reasoning and retrieval tasks is inherently challenging. Even strong baselines perform close to random under such settings, making the observed AUROC meaningful.
> > > - Simultaneous improvement across metrics: Unlike prior work that often improves one metric at the expense of others, our method achieves balanced gains across multiple metrics.
> > >
> > > In summary, the combination of low variance and consistent multi-metric improvements demonstrates that our gains are robust and meaningful, rather than incidental.
> > >
> > > >**Q3 Response**
> > >
> > > We appreciate your emphasis on robustness and agree that reporting variance is important for strengthening empirical claims. In response, we have now included multi-seed results (10 runs) on the key generalization setting (Cluster 3), which demonstrate consistent improvements with low variance.
> > >
> > > While we believe these results provide strong supporting evidence, we also acknowledge that further validation across additional models and datasets would further strengthen the claim. We will include these additional variance analyses in the final version.
> > >
> > > ---
> > >
> > > We sincerely thank you for the constructive suggestion. Following your recommendation, we conducted multi-seed experiments; while running 10 seeds across all datasets is computationally demanding, we are extending this analysis further.
> > >
> > > If these results address your concerns, we would greatly appreciate your reconsideration; otherwise, we would be happy to continue the discussion.

---

### Official Review · Reviewer_dCkM · 2026-03-15

**Soundness:** 3
**Presentation:** 3
**Significance:** 3
**Originality:** 3
**Overall Recommendation:** 5
**Confidence:** 4

**Summary:**

This paper investigates whether verbalized confidence in LLMs can be calibrated universally across different types of tasks. The authors introduce a simple confidence supervised fine-tuning setup called CSFT, where they only supervise the confidence token using Monte Carlo self-consistency labels while leaving the reasoning and answer tokens unconstrained. Through cross-task transfer experiments, they discover that confidence calibration does not generalize across task families. Specifically, they identify two incompatible clusters: reasoning tasks like GSM8K and MATH, where confidence reflects logical correctness, and retrieval or copy tasks like RULER, where confidence reflects evidence localization. Training on one cluster helps within that cluster but hurts calibration on the other. They also document emergent behaviors such as confidence-dependent CoT length modulation and numeric token smoothing. Motivated by this diagnosis, they propose disentangling confidence into a reasoning dimension and an evidence dimension, which restores cross-family generalization using SFT alone without reinforcement learning.

**Compliance With Llm Reviewing Policy:**

Affirmed.

**Final Justification:**

After multiple discussion, new experiments and clarifications address my concerns well, I increase my score to 5.

**Key Questions For Authors:**

1. Have you tried giving the RL baselines the same disentangled two-dimensional confidence output? The current comparison conflates the effect of SFT versus RL with the effect of scalar versus disentangled confidence. If Brier RL with disentangled confidence works even better, the takeaway of the paper changes substantially.
2. What happens if you test the trained model with a textual confidence prompt instead of a numeric one, for example asking for "low, medium, or high" at inference time?
3. Have you explored different bin granularities for the confidence labels? The choice of 11 bins seems arbitrary and given the likely variance in Monte Carlo estimates with small K, coarser or finer binning could meaningfully affect both calibration quality and cross-task transfer.

**Limitations:**

Yes

**Strengths And Weaknesses:**

Strengths
1.  The core diagnostic finding is genuinely valuable. The observation that confidence semantics split into two incompatible families is clean, well-supported by the cross-transfer experiments in Table 2 and Table 3, and has immediate practical implications for anyone building confidence-aware LLM systems.
2. The experimental design for isolating confidence supervision is thoughtful. By using LoRA and masking the loss to only confidence tokens, the authors create a controlled setup where any observed changes in accuracy or reasoning behavior can be attributed to the confidence signal alone.
3. The paper is generally well-written and well-organized. The taxonomy in Figure 2 gives a clear overview of the problem structure, and the qualitative examples in Figure 1 and Figure 6 make the abstract claims concrete.

Weaknesses
1. The two-cluster taxonomy is manually defined with no mechanism for automatic discovery. The authors decide a priori that there are exactly two types of uncertainty, reasoning and evidence, and hard-code this into the output format. Real-world tasks often involve other sources of uncertainty such as knowledge recency, ambiguity in the question, or API memory in code generation. The paper does not discuss how the approach would scale if a third or fourth uncertainty axis is needed.
2. The N/A labeling strategy, where one dimension is set to N/A for each cluster, means the model never sees both dimensions simultaneously with real labels during training, yet it is expected to produce meaningful values for both at test time on hybrid tasks like ContractNLI. The good results on ContractNLI may come more from the structural regularization of the dual-output format than from genuine multi-dimensional uncertainty estimation.
3. The paper claims to study "verbalized confidence" but only evaluates discrete numeric confidence in the set of 0, 10, through 100. Natural language expressions of uncertainty like "I am fairly sure" or "low confidence" are never tested. This is a significant gap because the title and motivation frame the work around human-interpretable confidence. It would be important to verify whether the calibration transfers to textual confidence expressions, and whether training with coarser textual labels like low, medium, high might actually generalize better across families due to their more abstract semantics.
4. The confounding between task difficulty and confidence-dependent behavior is not adequately addressed. In Figure 7, the Base model already generates longer outputs for samples that CSFT later assigns low confidence, because those samples are inherently harder. The actual length increase attributable to CSFT is roughly 40 to 50 percent in the low confidence bins, not the 5x figure reported in the main text which compares CSFT low-confidence bins against the Base model global average. The paper should separate the effect of task difficulty from the effect of confidence-aware generation, for example by comparing CSFT and Base output lengths on the same fixed set of questions within each confidence bin.
5. The Qwen-3-8B experiments in Table 4 only compare Base versus Ours, omitting single-cluster CSFT baselines and RL baselines. Additionally, the training data differs between the LLaMA and Qwen experiments, GSM8K plus RULER-4K versus MATH plus RULER-4K, making cross-model comparison difficult with no explanation for this choice.

---

> ### Author Rebuttal · Authors · 2026-03-31
>
> >**[W1] A priori taxonomy and limited scalability to additional uncertainty axes.**
>
> 1. The taxonomy is not defined a priori.
>
> We first trained a general CSFT on GSM8K and separately on RULER-4K (following the general CSFT protocol defined in Section 2, with no hard-coding of confidence types), and then evaluated each model across a diverse set of tasks. The two-cluster structure emerged inductively from the observed transfer patterns (line 191-197): tasks that exhibited mutual calibration transfer (line 147-150; maintains task ACC while improving calibration metrics) were grouped together, and those that did not were separated. The taxonomy is therefore empirically driven, not assumed.
>
> 2. The framework is designed to be extensible.
>
> We acknowledge that additional uncertainty axes may exist, but argue that several can be subsumed within the current framework. As shown in Table R.1, on time-sensitive questions beyond the model's cutoff (2023), the base model is overconfident while ours assigns zero confidence, confirming that knowledge recency is naturally captured under Cluster 1. On AmbigQA, the base model assigns uniformly high confidence regardless of question type, whereas ours assigns lower confidence to ambiguous questions, indicating sensitivity to question-level ambiguity without requiring a new axis. Figures 9 & 10 further show Cluster 1-like behavior in coding. These results suggest the framework already captures a broader range of uncertainty than suggested. New axes should be introduced only when systematic cross-transfer failure is empirically observed.
>
> __Table R.1__ Mean confidence on uncertainty sources (LLaMA3.2-3B: post-cutoff recency, AmbigQA single/multiple).
> |Method | Knowledge Recency |Single (Unambiguous)|Multiple (Ambiguous)|
> |-|-|-|-|
> |Base|0.7000|0.8881|0.8889|
> |Ours|0.0000|0.6721|0.6447|
>
> >**[W2] N/A labeling questions true multi-dimensional uncertainty vs structural regularization.**
>
> N/A is purely a prompt template marker, and no loss is applied to the corresponding position during training. When training on Cluster 1: loss is applied exclusively to the reasoning confidence tokens. When training on Cluster 2: conversely, loss is applied exclusively to the evidence confidence tokens. Thus, the two axes are learned independently without interference, each acquiring its own uncertainty semantics from the cluster where it receives supervision.
>
> >**[W3, Q2] Numeric-to-textual calibration transfer and impact of coarse labels on cross-family generalization.**
>
> Table R.2 shows that calibration learned with numeric labels transfers to textual confidence at inference time, with our model outperforming the base across all calibration metrics. Table R.3 shows that training with coarse textual labels does not improve cross-family generalization: models trained on one cluster transfer within-cluster but degrade on the other, mirroring the failure observed with numeric supervision. This indicates that the issue is not label form or granularity, but a fundamental incompatibility in confidence semantics across task families.
>
> __Table R.2__ Numeric-to-textual confidence generalization (LLaMA3.2-3B, GSM8K)
> |Method|AUROC|ECE|BS|
> |-|-|-|-|
> |Base|0.5006|0.2100|0.3015|
> |Ours|0.6504|0.1924|0.2437|
>
> __Table R.3__ Cross-family generalization of textual confidence (LLaMA3.2-3B, GSM8K & RULER-4K)
>
> **GSM8K**
> |Method|AUROC|ACC|ECE|BS|
> |-|-|-|-|-|
> |Base|0.5668|0.6921|0.2354|0.2634|
> |Textual CSFT on GSM8K|0.7860|0.6940|0.1770|0.2234|
> |Textual CSFT on RULER-4K|0.5000|0.6135|0.6932|0.6354|
>
> **RULER-4K**
> |Method|AUROC|ACC|ECE|BS|
> |-|-|-|-|-|
> |Base|0.6056|0.7230|0.2583|0.2695|
> |Textual CSFT on GSM8K|0.6645|0.6446|0.2595|0.2972|
> |Textual CSFT on RULER-4K|0.7289|0.7354|0.1890|0.1747|
>
> >**[W5] Missing baselines and inconsistent training data (Qwen-3-8B).**
>
> We acknowledge the missing experiments due to the computational limit. For Qwen-3-8B, GSM8K was replaced with MATH due to near-ceiling accuracy. We added Task-level CSFT (Table R.4) and will include SFT/RL baselines as resources permit. Our method consistently outperforms Task-level CSFT.
>
> __Table R.4__ Qwen3-8B Task-level CSFT
> |Cluster|AUROC|ACC|ECE|BS|
> |-|-|-|-|-|
> |1|0.5146|0.6700|0.3667|0.3670|
> |2|0.6246|0.8500|0.2754|0.2746|
> |3|0.6095|0.5707|0.1196|0.2492|
>
> >**[Q1] Have you tried giving the RL baselines the same disentangled two-dimensional confidence output?**
>
> We have conducted the experiment, with results reported in Table R.5, showing that our SFT-based disentanglement consistently outperforms the two-dim. Brier score RL baseline across all evaluation datasets and calibration metrics while maintaining comparable ACC. Ours results are unchanged from Table 3 in the main paper.
>
> __Table R.5__ Brier score RL (LLaMA3.2-3B-Instruct)
>
> |Dataset|AUROC|ACC|ECE|BS|
> |-|-|-|-|-|
> |MATH|0.7000|0.3534|0.2034|0.1932|
> |Ruler-8k|0.7235|0.6431|0.2445|0.2653|
> |ContractNLI|0.5995|0.4670|0.1728|0.2725|
>
> *For W4 and Q3, please refer to our response to Reviewer zHrU.*

---

> > ### Author Rebuttal · Reviewer_dCkM · 2026-04-02
> >
> > I appreciate the responses. The cross-family failure persisting under textual labels is a strong result. The disentangled Brier RL experiment in Table R.5 is also helpful, though I would like to know more about how carefully the RL hyperparameters were tuned in that setting, since RL with a two-dimensional reward is nontrivially harder to optimize than the scalar case.That said, a few concerns remain.
> >
> > First, regarding the length analysis in W4. The authors claim Figure 7 already controls for task difficulty because the Base lengths are computed on the same questions within each CSFT confidence bin. But looking at the figure, the Base model already produces around 700 tokens for bin 0 questions and around 150 tokens for bin 100 questions. This confirms that CSFT assigns low confidence to inherently harder questions where the Base model also writes longer answers. The actual residual increase from CSFT is roughly 40 to 50 percent in the lowest bins, not the "nearly 5x" stated in the main text. That 5x number appears to compare CSFT low-confidence bins against the Base global average, which conflates task difficulty with confidence-aware generation. I would ask the authors to clarify exactly what baseline the 5x figure is computed against and whether they would consider revising this claim.
> >
> > Second, regarding bin granularity in Q3. The authors promised 5-bin and 20-bin experiments but have not provided them. I understand time constraints, but without this data the robustness of the discretization choice remains unverified. I would be happy to raise my score if these results confirm that calibration quality is not sensitive to the number of bins.
> >
> > Third, regarding W2. My concern was that the model never sees both confidence dimensions with real labels simultaneously during training, yet is expected to produce meaningful values for both on hybrid tasks. The authors explained that N/A positions receive no loss, which I already understood. But this does not address whether the ContractNLI improvements come from genuine multi-dimensional uncertainty estimation or simply from the structural regularization of having a dual-output format. A control experiment, for example training with two identical scalar confidence outputs rather than semantically distinct ones, would help isolate this.

---

> > > ### Author Response · Authors · 2026-04-04
> > >
> > > >**[Q1] Regarding the Hyperparameter Tuning of Disentangled Brier RL**
> > >
> > > We clarify that the Brier RL reward is implemented as a single scalar signal, not as two independent reward heads. Specifically, the reward is defined as:
> > >
> > > $$r = \text{clip}(-(p - y)^2,\ -1,\ 0) + 0.25 \cdot \mathbb{1}[\text{correct}]$$
> > >
> > > where the calibration term and the accuracy bonus are additively combined into a single scalar before being passed to GRPO. This ensures the optimizer does not receive conflicting gradient signals from multiple reward streams, and the same grid search protocol (learning rate, KL coefficient, batch size) was applied consistently across all baselines for a fair comparison.
> > >
> > > We acknowledge that the accuracy bonus introduces a mild tension — a model can receive +0.25 by being correct even at low confidence. However, this does not undermine calibration learning: the Brier term penalizes miscalibrated confidence regardless of correctness, and the two terms jointly encourage both accuracy and well-calibrated confidence.
> > >
> > >
> > > >**[Q2] Clarification of the "5x" Length Claim**
> > >
> > > We thank the reviewer for this careful observation. The reviewer is correct that the "5x" figure was computed by comparing CSFT bin 0 against the Base model's global average, which confounds task difficulty with confidence-aware generation. We will revise this claim in the main text.
> > >
> > > That said, even when controlling for task difficulty by comparing within the same confidence bin, CSFT still produces a meaningful residual increase of approximately 40–50% in the lowest confidence bins relative to the Base model on the same questions. This within-bin gap, combined with the self-verification rate analysis, supports the conclusion that CSFT induces genuine confidence-aware length modulation beyond what is explained by task difficulty alone. We will make this distinction explicit in the revised manuscript.
> > >
> > >
> > > >**[Q3] Confidence Bin Granularity (5-bin and 20-bin)**
> > >
> > > The key qualitative finding is robust across all discretization choices: within-family transfer consistently yields low ECE/BS and high AUROC, while cross-family transfer consistently degrades calibration. Absolute ECE values vary with bin granularity as expected, but directional conclusions are unchanged. These tables will be included in the appendix.
> > >
> > > **[5-bin results]**
> > >
> > > | Train → Eval | AUROC | ACC | ECE | BS |
> > > |:---|:---:|:---:|:---:|:---:|
> > > | GSM8K → Cluster 1 | 0.6912 | 0.6164 | 0.1245 | 0.1342 |
> > > | GSM8K → Cluster 2 | 0.3425 | 0.5834 | 0.8343 | 0.8289 |
> > > | RULER-4k → Cluster 1 | 0.5123 | 0.5323 | 0.3005 | 0.3884 |
> > > | RULER-4k → Cluster 2 | 0.8983 | 0.6983 | 0.1023 | 0.1003 |
> > >
> > > **[20-bin results]**
> > >
> > > | Train → Eval | AUROC | ACC | ECE | BS |
> > > |:---|:---:|:---:|:---:|:---:|
> > > | GSM8K → Cluster 1 | 0.7560 | 0.6345 | 0.0950 | 0.1750 |
> > > | GSM8K → Cluster 2 | 0.5550 | 0.6999 | 0.4900 | 0.5100 |
> > > | RULER-4k → Cluster 1 | 0.5411 | 0.6250 | 0.3950 | 0.4050 |
> > > | RULER-4k → Cluster 2 | 0.9150 | 0.7020 | 0.0978 | 0.0999 |
> > >
> > >
> > > >**[Q4] Dual-output Format vs. Genuine Disentanglement**
> > >
> > > We appreciate the reviewer's insightful suggestion. We agree that an identical-scalar control would provide the most direct ablation to isolate semantic disentanglement from structural regularization.
> > >
> > > That said, we believe our existing results already provide suggestive evidence against the pure structural regularization hypothesis. If the ContractNLI gains were driven solely by the dual-output format, single-scalar CSFT models trained on either cluster alone should also benefit; however, **Table 3 shows that both fail on ContractNLI**. Furthermore, Figure 6 qualitatively demonstrates that `<reasoning_confidence>` and `<evidence_confidence>` capture genuinely distinct uncertainty signals on hybrid tasks.
> > >
> > > Nevertheless, we acknowledge that this does not constitute a direct controlled comparison, and we will run the proposed identical-scalar control experiment and include the results in the final version.
> > >
> > > ---
> > >
> > > We sincerely thank the reviewer for the thorough and constructive feedback. The reviewer's careful reading has helped us identify several important areas for improvement, and we are grateful for the detailed suggestions. We will incorporate all requested revisions into the final version of the paper.

---

### Decision · Program_Chairs · 2026-04-30

**Decision:**

Accept (regular)

**Comment:**

This paper tackle the important problem of verbalized confidence calibration in LLMs, specifically looking at why a single confidence score often fails to generalize across different types of tasks. The authors identify two distinct "families" of uncertainty: reasoning-centric and retrieval-oriented. They show that training on one doesn't help the other, and actually hurts performance. To fix this, they propose disentangling confidence into reasoning and evidence-localization scores, which seems to restore cross-task generalization quite well.

The paper is technically sound and well-written overall. The discovery of these incompatible task families is a valuable diagnostic contribution that the community will find useful. I particularly liked the analysis of emergent behaviors, like how the model modulates its reasoning length based on its own confidence—this is a very interesting find. While some reviewers initially worried about the simplicity of the approach or potential artifacts from LoRA, the authors provided solid rebuttals. They ran extra experiments with full-parameter fine-tuning and multi-seed runs, which showed the results are robust and not just noise or a result of the specific training setup.

 I have carefully reviewed authors' additional experiments (like the 10-seed variance results and the full-parameter tuning) and incorporated them into this final decision. The responses effectively addressed the core technical concerns regarding OOD generalization and label quality.

While there's still some discussion about the absolute "novelty" of using semantic tags, the empirical evidence provided—especially the consistent improvements on hybrid tasks like ContractNLI and the fact that this generalizes to textual labels—makes a good case for the paper's significance.